# From single scenes to extended scenarios: The role of the ventromedial prefrontal cortex in the construction of imagery-rich events

Julia Taube[1,2☯], Pitshaporn Leelaarporn[1,2☯], Maren Bilzer[1,2,3], Rüdiger Stirnberg[2], Yilmaz Sagik[2,4], Cornelia McCormick[1,2]*

1 University of Bonn, University Hospital Bonn, Clinic for Geriatric Psychiatry and Cognitive Disorders; Bonn, Germany, 2 German Center for Neurodegenerative Diseases (DZNE), Bonn, Germany, 3 University Bonn, Bonn, Germany, 4 Department of Neurosurgery, University Hospital Tuebingen, Tuebingen, Germany

☯ These authors contributed equally to this work.
* cornelia.mccormick@ukbonn.de

## Abstract

Mental events are fundamental to daily cognition, including the recollection of past experiences, the anticipation of future scenarios, and engagement in imaginative, fictitious thought. Typically, these temporally extended mental events unfold within coherent spatial contexts, rich in naturalistic scenes and objects. However, there remains a significant gap in understanding how these events are represented in the brain. This study aimed to investigate the neural patterns involved in the construction of temporally extended mental events. Using ultra-high field functional magnetic resonance imaging, we examined brain regions previously implicated in this cognitive process, including the ventromedial prefrontal cortex (vmPFC), hippocampus, and posterior neocortex. We employed a novel experimental paradigm in which participants engaged in three forms of mental imagery: single objects (e.g., "a black espresso"), single scenes (e.g., "a busy café"), and extended scenarios (e.g., "meeting a friend for coffee"). We identified a shared neural network, comprising the vmPFC, hippocampus, and posterior neocortex, engaged across all forms of mental imagery. However, we observed a hierarchical organization in their contributions: the posterior neocortex supported the construction of objects, scenes, and scenarios, while the hippocampus primarily contributed to scenes and scenarios. The vmPFC exhibited a stepwise increase in activation, peaking during scenario construction. These findings suggest that the construction of mental events involves dynamic interactions between perceptual representations in the posterior neocortex, spatial coherence provided by the hippocampus, and integrative processes within the vmPFC. While the vmPFC may play a particularly prominent role in constructing temporally extended scenarios, it likely also contributes to the integration of elements within single scenes.

**Data availability statement:** The data analyzed in this study were collected as part of a larger clinical investigation involving a pilot cohort of healthy participants. Due to the conditions specified in the informed consent process, the dataset cannot be made publicly available. Qualified researchers who meet the criteria for access to confidential data may submit requests to the authors of this manuscript (Cornelia.mccormick@ukbonn.de). All requests will be reviewed according to the institution's data sharing protocols and ethical guidelines. A detailed description of the variables and the data structure can be provided upon reasonable request, subject to institutional review board approval and data sharing agreements. The data protection office at the University Hospital Bonn is another point of contact to request access to the data: Achim Flender (Achim.Flender@ukb.uni-bonn.de), Venusberg-Campus 1,53127 Bonn, Germany.

**Funding:** This research was supported by the Hertie Network of Excellence in Clinical Neuroscience. Work in C.M.'s lab is further financed by internal research funding of the Faculty of Medicine (BONFOR), University Hospital Bonn, by the Federal Ministry of Education and Research (BMBF) within the framework of the funding programme ACCENT (funding code 01EO2107) and by the Deutsche Forschungsgemeinschaft (DFG, German Research Foundation, MC 244/3-1). J.T. received an Argelander Mobility Grant from the University Bonn. The funders had no role in study design, data collection and analysis, decision to publish, or preparation of the manuscript.

**Competing interests:** The authors have declared that no competing interests exist.

## Introduction

Mental events are fundamental to everyday cognition, enabling the recollection of autobiographical events, the envisioning of our future selves, and the creation of imagined scenarios. For most individuals, these movie-like mental events unfold within a coherent spatial framework, where naturalistic scenes and objects serve as key features. While extensive research has linked the construction of naturalistic scenes to the hippocampus [1–6], the specific contributions of the ventromedial prefrontal cortex (vmPFC), and posterior neocortex in constructing these temporally extended mental movies remain less understood. To distinguish terminology, we use "mental events" as a broader term for all types of mental activity including autobiographical memory recall, future thinking, etc.

We have recently suggested that the vmPFC plays a pivotal role that goes beyond mentally constructing individual objects and scenes [7]. Within this framework, the vmPFC contributes not only to scene construction but is particularly critical for initiating and elaborating temporally extended, imagery-rich mental events, hereafter referred to as "scenarios". This view emerged from a detailed examination of the cognitive effects of bilateral hippocampal damage, and bilateral vmPFC lesions.

First, previous research described that the spontaneous initiation of internal mental events (e.g., mind-wandering episodes) appeared not only reduced in vmPFC-damaged patients [8], but also their off-task thoughts related less to the future and more to the present in comparison to controls. Importantly, the presence of hippocampal lesions did not lead to a measurable decrease of mind-wandering episodes, but their episodes appeared less visual and verbal semantic in nature [9–11].

Second, hippocampal-damaged patients have consistently been shown to report spatially fragmented scenes or autobiographical episodes containing fewer sensory or episodic details [12–14]. Findings regarding the effects of vmPFC damage, however, are more mixed. Kurczek et al. (2015) reported that patients with vmPFC lesions were able to recall and describe coherent individual scenes [15], whereas De Luca at el. (2018) found that such patients provided fewer spatial and other descriptive details when constructing complex scenes [16]. The authors also leveraged boundary extension, i.e., a cognitive process, in which viewers, after seeing a scene, automatically construct an internal representation that extends beyond the actual visual boundaries, leading to later memory for more than was shown, and found that both hippocampal and vmPFC lesions lead to impairments [16]. Together, these findings suggest that while the vmPFC may not be primarily responsible for the visuo-spatial or mnemonic binding of scene elements, it likely supports integrative and organizational processes required for the temporal unfolding of dynamic, scene-based mental scenarios [17]. Consistent with this view, a meta-analysis found that the vmPFC was more strongly associated with the term "event" than with "scene" [18].

We propose that the vmPFC initiates the activation of specific mental representations, particularly those with temporal and spatial extendedness, and conveys this information to the hippocampus, which constructs individual scene snapshots from the broader scenario [19]. The vmPFC then engages in iterative feedback loops with

the hippocampus and posterior neocortex, progressively integrating successive scenes and perceptual details into a temporally unfolding mental scenario.

Several magnetoencephalography (MEG) studies provide strong support for this model, demonstrating that the vmPFC plays a pivotal role in driving hippocampal scene construction processes [20–23]. McCormick et al. (2020) demonstrated that the vmPFC responds faster than the hippocampus during the initiation of autobiographical memory retrieval driving hippocampal evoked responses. Monk et al. (2020) further elucidated this interaction by showing that the vmPFC drives activity in the hippocampus during scene construction. Monk et al. (2021) expanded on these findings by demonstrating that the same interaction between the vmPFC and hippocampus occurs during tasks where scenes are integrated into the unfolding of a broader scenario.

Scene Construction Theory posits that the hippocampus plays a central role in generating spatially coherent mental representations by integrating disparate elements into a unified scene. This process involves the anterior hippocampus, which is thought to support the generation of novel spatial contexts and the binding of multimodal information, such as spatial, temporal, and sensory details, into a cohesive framework [24–27]. Previous research specifically connects scene construction to the hippocampus, with evidence highlighting the significant involvement of its anterior segment [3–5,7,12,28].

Thus, the current fMRI study investigates the precise contributions of the vmPFC, hippocampus, and posterior neocortex to the construction of naturalistic temporally extended mental imagery-rich scenarios. Using 7T MRI, we were able to achieve unprecedented spatial resolution and signal-to-noise ratio, allowing for more precise mapping of neural activity within our key regions. This level of detail is particularly crucial for understanding scenario construction, as it enables (1) the identification of subtle activation differences that may be obscured at lower field strengths and (2) the detection of fine-grained connectivity patterns. By leveraging these advantages, our study provides novel insights into the hierarchical organization of brain regions involved in constructing mental imagery-rich events.

Our hypotheses were as follows: (1) We expect the vmPFC to exhibit greater activation during scenario construction compared to scene and object construction. (2) We expect the hippocampus, particularly the anterior segment, to be more engaged during scenario and scene construction than during object construction. (3) We expect the posterior neocortex to be involved in all three types of imagery.

To test these hypotheses, we implemented a multimodal design integrating behavioral, eye-tracking, and ultra–high-field 7 T fMRI data acquisition. The eye-tracking experiment served both methodological and theoretical purposes. Methodologically, it verified that participants understood and adhered to the imagery instructions. Theoretically, eye movements offer a sensitive behavioral index of internally generated scene and scenario construction, as oculomotor patterns reflect the vividness, spatial coherence, and task-complexity of mental representations [29–31].

Participants subsequently performed the same task during 7 T fMRI to precisely localize neural activity and connectivity associated with constructing temporally extended scenarios. A non-word counting task served as a control condition to dissociate constructive imagery from general cognitive demands. Finally, an unexpected post-scan source memory test assessed recognition and contextual recall of stimuli. This integrative approach allowed us to characterize the behavioral, oculomotor, and neural mechanisms underlying mental event construction and to delineate the specific contributions of the vmPFC, hippocampus, and posterior neocortex.

## Materials and methods

### Participants

Twenty-two healthy, right-handed participants were initially enrolled in the study after providing oral and written informed consent between 05.12.2022 and 30.10.2023. Three participants were excluded due to suboptimal performance in the classification task during scanning (i.e., they performed in the 4th percentile in at least one imagery category). The final sample comprised of nineteen participants (11 female, 8 male) with no history of neurological or psychiatric disorders and a mean age of 27.89 ± 3.67 years.

For our experiment, it was crucial that participants reported a typical ability to visualize mental content. To assess this, we administered a German version of the Vividness of Visual Imagery Questionnaire (VVIQ), a 16-item scale that asks respondents to internally visualize four different scenarios and rate the vividness of their mental images on a 5-point Likert scale, yielding a total subjective vividness score [32]. Our exclusion criterion was a VVIQ score below 32, to exclude individuals with aphantasia [33]. No participants were excluded based on their VVIQ score. The mean score of our participants was 58.63 ± 10.39, indicating that they showed typical imagery ability.

To control for depressive symptoms, we included the German Beck Depression Inventory V (BDI-V), which assesses depressive symptoms across 21 items, each rated on a 6-point Likert scale, with a cut off score of 35 [34]. Mean scores were 20.78 ± 8.51. No participants were excluded based on their BDI-V score. This study was reviewed and approved by the local ethics board of the University Hospital Bonn, Germany (Proposal 383/22). The study has been pre-registered to osf.io (https://doi.org/10.17605/OSF.IO/C46YR).

## Stimuli selection

The stimuli comprised four types of verbal cues eliciting mental representations of objects, scenes, scenarios; or a non-word (a control task not involving visual construction). Object and scene stimuli were selected from a previous study on imageability of words [28] and translated into German. For the scenario condition, we designed novel verbal cues.

Before the main experiment, an independent sample of 23 healthy participants rated the type of imagery elicited by 165 verbal cues in an online questionnaire. Participants were first instructed to visualize each cue as vividly and in as much detail as possible, without being told about any specific imagery condition. They were then asked to indicate which type of imagery the cue elicited. Critically, participants were not told about any experimental imagery conditions, nor were they instructed to imagine differently according to category. They simply formed their own spontaneous image for each cue, then labelled the image type. Based on these ratings, we selected cues using a discrimination-accuracy criterion: only cues that consistently elicited a single imagery category above a prespecified threshold were retained.

For the fMRI experiment, we selected only those stimuli that demonstrated the highest categorical discrimination (>80% accuracy), maximizing the reliability of neural responses during scanning. Stimuli with moderate discrimination rates (>60% accuracy) were allocated to the eye-tracking experiment. The remaining stimuli, which showed the lowest discrimination values while still reaching accuracy levels above chance, served as lures in the source memory task after scanning.

Based on classification accuracy values, stimuli in the fMRI experiment reached 87% (SD = 5%) for objects, 86% (SD = 4%) for scenes, and 88% (SD = 5%) for scenarios. In the eye-tracking experiment, accuracy was 73% (SD = 5%) for objects, 72% (SD = 6%) for scenes, and 73% (SD = 6%) for scenarios. For the source memory task, accuracy was 75% (SD = 4%) for objects, 63% (SD = 5%) for scenes, and 73% (SD = 6%) for scenarios.

Stimuli of each imagery type (i.e., object, scene, scenario, and non-words) were matched for word length. In the eye-tracking task, average word lengths were 10.95 (SD = 5.60) for objects, 13.00 (SD = 7) for scenes, and 11.00 (SD = 4.21) for scenarios. In the fMRI task, means were 11.95 (SD = 4.97) for objects, 12.45 (SD = 6.14) for scenes, 14.25 (SD = 6.02) for scenarios, and 11.65 (SD = 4.15) for non-words. In the source memory task, means were 13.65 (SD = 5.78) for objects, 13.40 (SD = 8.02) for scenes, and 11.45 (SD = 4.73) for scenarios. Word length did not differ significantly between the conditions (p's > .05).

## General experimental procedure

Participants were first given verbal explanations and concrete examples of objects, scenes, and scenarios. They then practiced several trials in which they generated imagery from cues. After demonstrating comprehension, participants completed a computer-based practice session. In this session, they followed the instruction: "Please visualize the following cue as vividly and in as much detail as possible or count the letters in case of a non-word." Afterwards, they indicated

whether they had imagined an object, a scene, or a scenario, or whether they had counted letters. They were then introduced to the vividness rating scale: "How vivid was the imagined content? Please rate from very vague to very vivid.". Once they answered correctly and had no further questions, the eye-tracking session was conducted. This task served both as an extended practice and as a final check that participants fully understood the imagery instructions before entering the MRI scanner. After the MRI session, participants completed a source memory task.

## Imagery task during eye-tracking and fMRI

During the eye-tracking and fMRI experiment, participants performed three mental imagery tasks (object, scene, or scenario construction) and a control task. On imagery trials, they were instructed to internally construct and maintain a vivid mental image of the cued content while avoiding reliance on autobiographical memories. Importantly, participants were not informed of the imagery category associated with any cue. They formed a spontaneous image for each cue and only subsequently labeled the imagery type it elicited.

Participants were instructed to label the verbal cue as an object (e.g., a black espresso), if a single, detailed version of the object in isolation, akin to a catalogue photo (e.g., a shiny, black espresso in a small red cup with a fish-scale pattern and a green handle) was formed in their mind. It was explained to them that imagining scenes (e.g., a busy café), would be characterized by a spatially coherent, static image integrating multiple elements, resembling a postcard-like view (e.g., a street café in a picturesque Italian village with guests sitting at round tables under blue umbrellas, waiters carrying trays with cups and plates and colourful houses in the background).

To distinguish scenes from scenarios, participants made the categorization themselves based on the presence or absence of implied motion. When no motion was implied, the image was classified as a scene. When motion was implied (e.g., waiters serving guests), participants were instructed to either (a) mentally 'freeze' the image and focus on it as a static, cohesive scene, or (b) treat it as scenario imagery (e.g., meeting a friend for coffee), in which they constructed a dynamic mental simulation unfolding over time before their mind's eye. For example, meeting a friend for coffee could involve greeting them at a busy train station, walking to a café, and catching up over coffee.

Scenario construction differed from object and scene imagery by incorporating temporally extended scenarios with multiple scenes and objects. While all three conditions required mental imagery, they varied in their demands on scene construction and temporal complexity.

During the control task (only in the fMRI experiment), participants were required to count the number of characters in meaningless letter strings, matched in length to the experimental cues. The non-word counting task was chosen as a control condition because it requires basic cognitive engagement (e.g., attention and working memory) without involving mental imagery or constructive processes. This control condition allowed us to observe differences in brain activation attributable to imagery construction rather than general cognitive demands.

## Eye-tracking experiment

Eye movements were recorded using a video-based eye tracker (EyeLink 1000, SR Research) at a sampling rate of 2000 Hz. The system tracked participants' eyes while their head position was stabilized using a chin rest positioned 64 cm from the 47 cm wide screen display and 58 cm from the desktop-mounted camera. Before the experiment started, a 9-point calibration and validation procedure were performed to ensure accurate tracking (average error < 0.5° of visual angle).

During the eye-tracking task, participants were shown 60 pseudo-randomized stimuli across one session, containing 20 scenario, 20 scene, and 20 object stimuli. Each trial began with a one-second fixation cross, followed by a one-second stimulus and a five-second blank display. During these five seconds, they were instructed to visualize the stimuli as detailed as possible in their mind's eye without relying on past experiences. The stimulus was presented on a grey background to avoid afterimages in the blank display. This was followed by a (1) classification task, where participants categorized their imagined content freely as an object, scene, or scenario, and a (2) vividness rating, where they

rated their perceived vividness on a 4-point Likert scale, ranging from very vague to very vivid. Participants responded in a self-paced manner (up to a maximum of three seconds). The experiment was run using SR Research Experiment Builder v. 1.1.

### MRI experiment

**MR image acquisition.** Structural and functional MRI data were acquired using a MAGNETOM 7 T Plus ultra-high field scanner (Siemens Healthineers, Erlangen, Germany).

As previously described [27], a whole-brain T1-weighted multi-echo MPRAGE scan with 0.6 mm isotropic resolution was acquired using a custom sequence optimized for scanning efficiency and minimal geometric distortions [35,36]. The scan parameters were TI = 1.1 s, TR = 2.5 s, TEs = 1.84/3.55/5.26/6.92 ms, FA = 7°, TA = 7:12, readout pixel bandwidth: 970 Hz, matrix size: 428 x 364 x 256, elliptical sampling, sagittal slice orientation, CAIPIRINHA 1 x $2_{z1}$ parallel imaging with online 2D GRAPPA reconstruction, and a turbofactor of 218. The four echo time images were combined into a single high-SNR image using a root-mean-squares combination.

For functional imaging, a custom interleaved multishot 3D echo planar imaging (EPI) sequence was used with the following parameters: TE = 21.6 ms, $TR_{vol}$ = 3.4 s, FA = 15°, 6/8 partial Fourier sampling, oblique-axial slice orientation along the anterior-posterior commissure line, readout pixel bandwidth: 1136 Hz, matrix size: 220 x 220 x 140. This sequence achieved a high spatial resolution of 0.9 mm isotropic at 7 T with sufficient SNR and a BOLD-optimal TE by combining several features: (A) Skipped-CAIPI 3.1 x $7_{z2}$ sampling [37] with online 2D GRAPPA reconstruction, (B) one externally acquired phase correction scan per volume, (C) variable echo train lengths with a semi-elliptical k-space mask [38], and (D) rapid slab-selective binomial-121 water excitation. A 3-min fMRI practice run was followed by two main functional sessions (~15 min each). A standard 3 mm isotropic two-echo gradient-echo field-mapping scan was acquired in 35 s. A maximum of 264 imaging volumes were acquired per session, excluding the first five images to avoid non-steady-state signals.

**fMRI visual imagery task.** Directly prior to the scanning procedure, participants underwent a short practice to get accustomed to the response boxes. During scanning, participants were shown 60 pseudo-randomized stimuli across two sessions, containing 15 object, 15 scene, 15 scenario, and 15 non-word stimuli. They were instructed to visualize the stimuli (without relying on past experiences) or count the letters for non-words. Each stimulus was displayed for 10 seconds, and participants were instructed to engage in the respective imagery task throughout the entire presentation period.

This was followed by a self-paced classification task, during which participants categorized their imagined content as an object, scene, scenario, or whether they were counting. On a trial-by-trial basis, they also rated their perceived vividness on a 4-point Likert scale, ranging from very vague to very vivid, via keypress. Participants responded in a self-paced manner (up to a maximum of 5 seconds).

The inter-stimulus interval ranged from 1 to 4 seconds. The experiment was run using Cogent2000 version 125 (Wellcome Centre for Human Neuroimaging, UCL, London, UK).

**Source memory task: post fMRI.** After scanning, participants underwent an unexpected memory test. The stimuli presented included those previously shown during the eye-tracking and scanning sessions (old), as well as novel stimuli. The memory task consisted of 60 pseudo-randomized trials, with 20 trials per imagery condition (object, scene, scenario). Each condition included 10 novel and 10 old stimuli. Of the 10 old stimuli, 5 stimuli were taken from the eye-tracking, and 5 from the scanning experiment. Participants were required to indicate whether they had encountered each stimulus before and, if so, specify the context (eye-tracking or scanning) in which it was seen, or identify it as new.

**Debriefing question.** We also asked participants to describe their strategies for visualizing objects, scenes, and scenarios, and whether these strategies differed between imagery types and to which degree their constructions resembled autobiographical memories.

## Data analysis

**Behavioural data.** Behavioural data were analysed using separate repeated-measures (rm) ANOVAs, with the experimental condition (object, scene, scenario, non-word) as the within-subjects factor. For data collected during eye-tracking and scanning, we included classification accuracy, vividness ratings and reaction times as dependent variables. Reaction times were included as an indirect measure of the cognitive ease with which participants can classify a verbal cue. RTs can thus serve as a proxy for the fluency with which participants generate and differentiate between object, scene, and scenario imagery type.

For data collected after scanning, we included source (eye-tracking, scan, novel) as a second within-subjects factor and source memory accuracy (e.g., correctly recognizing a stimulus as the pre-defined category from the respective correct session) as the dependent variable.

Post-hoc comparisons were performed using paired t-tests with Bonferroni correction for multiple comparisons. Effect sizes were reported using partial eta-squared ($\eta^2 p$) for ANOVAs. The debriefing questions were analysed qualitatively.

Data analysis was conducted via Python libraries, including pandas, numpy, seaborn, matplotlib and pingouin. All planned contrasts were performed. For transparency, we report in the Results section all statistically significant findings. Non-significant contrasts are not detailed to avoid overloading the text, but were conducted and are available upon request.

**Eye-tracking data analysis.** For the five-second imagination duration, we analysed four eye movement parameters: (1) fixation count, (2) average fixation duration, (3) saccade count, and (4) average saccade amplitude. For eye-tracking data analysis, individual trials identified as outliers (beyond 3 standard deviations from the mean) were removed on a trial-by-trial basis, with 0–22 data points removed per parameter across all participants. No significant differences were observed between imagery categories in the outlier analysis, to ensure the missing data represented random outliers rather than systematic patterns. Through this approach the remaining data maintained sufficient power for repeated-measures ANOVA analysis with all 19 participants retained in the analysis.

Statistical analyses were conducted using rm ANOVAs for each eye movement parameter, with imagery category (object, scene, scenario) as the within-subjects factor. Post-hoc comparisons were performed using paired t-tests with Bonferroni correction for multiple comparisons. Effect sizes were reported using partial eta-squared ($\eta^2 p$) for ANOVAs. Data analysis was conducted via Python libraries, including pandas, numpy, seaborn, matplotlib and pingouin. All planned contrasts were performed. For transparency, we report in the Results section all statistically significant findings. Non-significant contrasts are not detailed to avoid overloading the text but were conducted and are available upon request.

**MRI preprocessing.** Preprocessing of MRI data was conducted using the SPM12 (Statistical Parametric Mapping 12) software package (Wellcome Trust Centre for Neuroimaging, London, UK; www.fil.ion.ucl.ac.uk/spm/) implemented in MATLAB R2019b (MathWorks, Natick, MA, USA). First, anatomical and functional images were reoriented to align with the anterior-posterior commissure axis, ensuring standardized orientation across all subjects. Field maps, comprising phase and magnitude images, were utilized to compute voxel displacement maps (VDMs). These VDMs were subsequently applied during the realignment and unwarping process to correct for geometric distortions in the echo-planar imaging (EPI) sequences. The mean functional image was co-registered to the high-resolution anatomical image. The subject's anatomical image was segmented and spatially normalized to the T1-weighted Montreal Neurological Institute (MNI) template. Normalization parameters were then written to the functional data using the deformation fields derived from the structural image. For smoothing, a Gaussian kernel of 6 mm FWHM was applied, balancing between spatial specificity and group-level sensitivity. Rigid-body realignment was performed on the functional data to correct for head movement during scanning.

**Partial least squares.** The pre-processed fMRI data were analysed applying a partial least squares (PLS) approach, a covariance- based multivariate analysis technique with the advantage that there are no assumptions about the shape of the hemodynamic response function [39,40]. PLS employs singular value decomposition (SVD) to

derive ranked latent variables (LVs) from the covariance matrix of brain activity and experimental conditions. These LVs represent patterns of brain activity that are associated with each experimental condition. Statistical significance of the LVs was assessed using permutation testing (n = 500) and a value of $p < .05$ was considered significant. The reliability of each voxel contributing to the LV was assessed by bootstrapping (n = 100) resulting in bootstrap ratios (BSRs). Clusters of 10 or more voxels with a $BSR < 2.5$ (for Task-based PLS) and $BSR < 2.0$ (for Seed PLS) were considered reliable, resembling approximately $p < .05$.

Since we were interested in activity and connectivity differences between conditions, we employed a two-stage approach. In the first step, we applied a data-driven approach (mean-centred task-based PLS) to identify distinct patterns of brain activity during (1) imagery and control trials, as well as (2) in the imagery tasks only.

In the second step, we used Seed PLS analysis to assess differences in functional connectivity of the vmPFC. Seed PLS analysis investigates the relationship between the signal intensities of a predefined target region (seed voxel = vmPFC) and those of all other brain voxels, considering the influence of experimental conditions. This multivariate technique identifies patterns of brain activity that are maximally correlated with the seed region's activity, thereby elucidating functional connectivity and its modulation by experimental manipulations. Importantly, positive or negative correlations do not mean increased or decreased brain activity. The sign of the brain score/ correlation reflects the polarity of a region's contribution to the latent variable rather than the direction of activation change. We conducted one seed PLS analysis including the three imagery tasks (mean-centred task-based Seed PLS).

**Signal intensity extraction.** Since episodic future simulation and episodic recall rely on similar brain regions, we extracted signal intensities from regions typically involved in autobiographical memory. The coordinates were chosen via meta-analyses maps created with Neurosynth (https://neurosynth.org/) using the term "autobiographical memory". We included the vmPFC (MNI −4 54–12), left/right anterior hippocampus (l: −24 −22 −18, r: 24 −14 −16), left/right posterior hippocampus (l: −28 −34 −8, r: 30 −36 −8), left/right parahippocampal gyrus (l: −24 −40 −18, r: 32 −38 −14), retrosplenial cortex (−4–58 20), and left/right visual cortex (l: −44 −72 −2, r: 46 −72 −4). Of note, positive or negative signal intensity values extracted from PLS do not reflect fMRI activation or deactivation.

## Results

### Eye-tracking

The following paragraphs include the behavioural and eye movement results from the eye-tracking experiment.

**Classification task.** Classification accuracy did not differ between imagery type ($F(2,36) = 3.07$, $p = .06$, $\eta^2p = .15$). Scenarios (M = 84%, SD = 37%), scenes (M = 75%, SD = 43%) and objects (M = 72%, SD = 45%) were accurately classified in the majority of trials. In cases of miss-classification, scenarios and objects were most frequently misclassified as scenes, occurring in 14% and 17% of the trials, respectively. To clarify, the eye-tracking task was conducted before scanning and served primarily as a behavioural validation and training session, confirming that participants could reliably generate and distinguish object, scene, and scenario imagery.

**Vividness ratings.** Mean vividness ratings did not differ significantly across imagery conditions, $F(2, 36) = 0.89$, $p = .419$, $\eta^2p = .025$, indicating comparable levels of vividness for scenarios (M = 1.96, SD = 0.73), objects (M = 2.07, SD = 0.79), and scenes (M = 2.06, SD = 0.75).

**Reaction times.** We found significant main effects of the imagery category on reaction times in the classification task ($F(2,36) = 11.04$, $p < .001$, $\eta^2p = .076$). Post-hoc tests showed that scenarios (M = 1353ms) were processed significantly faster than both objects (M = 1578ms; $p = .024$) and scenes (M = 1717ms; $p = .001$). The shorter reaction times for scenarios compared to scenes and objects suggest that participants could more readily generate or classify temporally extended, event-like imagery during eye-tracking.

**Eye movements.** Scene construction elicited the highest fixation count, with a significant main effect of imagery category, $F(2, 36) = 8.88$, $p < .001$, $\eta^2_p = .33$ (see Fig 1). Post-hoc tests showed that fixation count was significantly greater

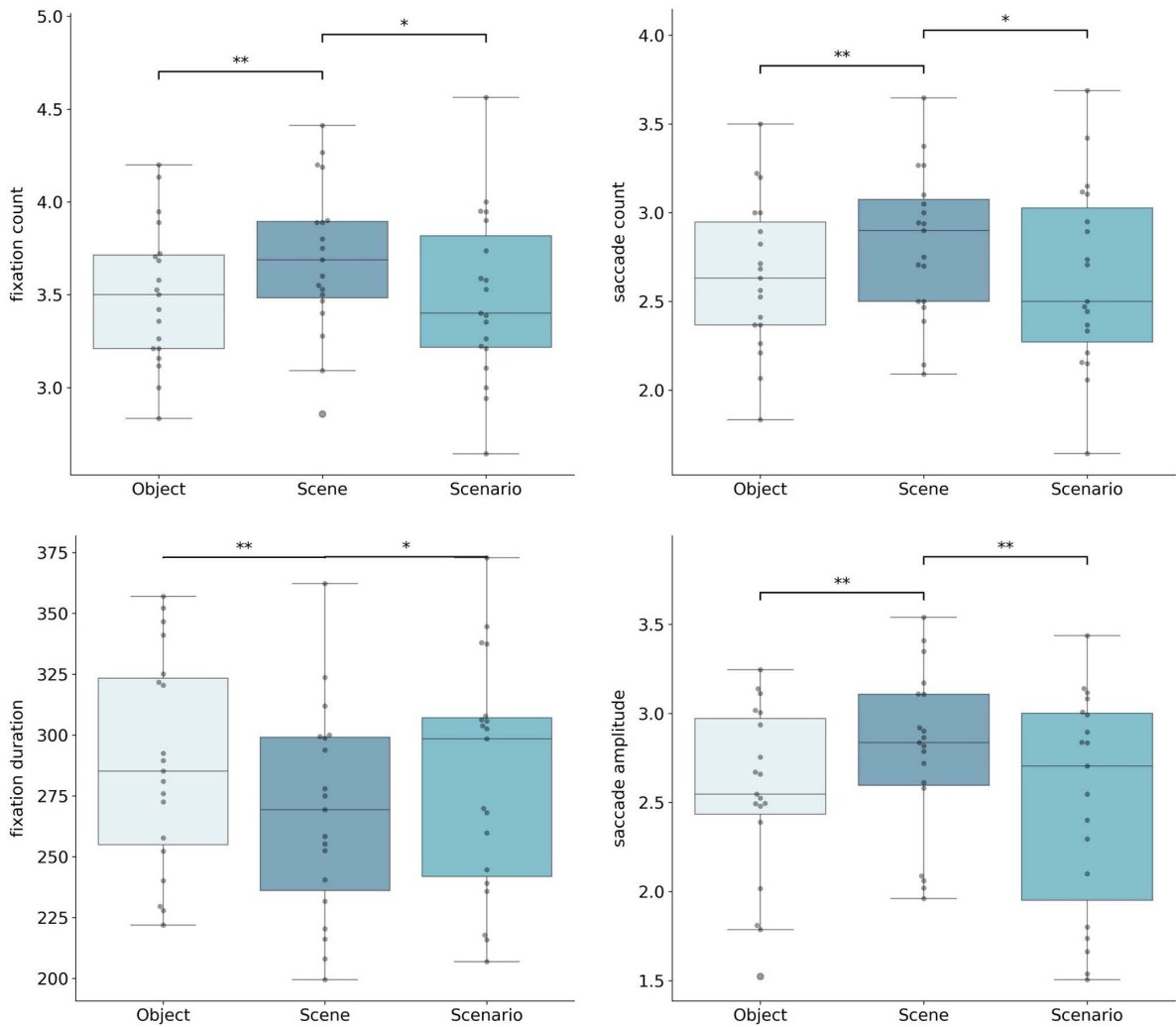

**Fig 1. Eye movement data.** Bargraphs display fixation counts/ duration, saccade count/ amplitude during the imagery trials *p < .05, **p < .01, ***p < .001.

for scenes than for both scenarios (*p* = .01) and objects (*p* = .006). Mean fixation counts were 3.72 (SD = 0.97) for scenes, 3.52 (SD = 0.97) for objects, and 3.50 (SD = 0.93) for scenarios.

Average fixation duration was shortest for scenes, with a significant main effect of imagery category, $F(2, 36) = 10.79$, $p < .001$, $\eta^2_p = .32$. Post-hoc tests indicated that fixation duration for scenes was shorter than for both objects (*p* = .004) and scenarios (*p* = .01). Mean durations were 266.93 ms (SD = 99.44) for scenes, 287.63 ms (SD = 107.55) for objects, and 282.03 ms (SD = 102.78) for scenarios.

Scene imagery also elicited more frequent saccades and larger amplitudes compared to both scenario and object imagery. Significant main effects were observed for saccade count, $F(2, 36) = 6.01$, $p = .006$, $\eta^2_p = .25$, and saccade amplitude, $F(2, 36) = 7.15$, $p = .002$, $\eta^2_p = .28$ (see Fig 1). Post-hoc comparisons revealed higher saccade counts and amplitudes for scenes relative to both scenarios and objects (*p* < .05). Mean saccade counts were 2.85 (SD = 1.03) for scenes, 2.65 (SD = 0.99) for scenarios, and 2.67 (SD = 1.01) for objects; mean amplitudes were 2.84° (SD = 1.73) for scenes, 2.55° (SD = 1.44) for scenarios, and 2.59° (SD = 1.50) for objects. [4,41,42].

## Scanning task

The following paragraphs include a description of the behavioural results during the fMRI task and the debriefing session.

**Classification task.** In the subsequent scanning session, classification accuracy showed a significant main effect of imagery category ($F(3,54) = 3.41$, $p < .05$, $\eta^2_p = .16$), but no significant post-hoc differences between conditions (non-words: $M = 91\%$, $SD = 16\%$; objects: $M = 90\%$, $SD = 9\%$; scenarios: $M = 83\%$, $SD = 10\%$; scenes: $M = 79\%$, $SD = 21\%$; see Fig 2a). As in the eye-tracking task, scenes were most frequently confused with scenarios (16% of trials), while objects were occasionally classified as either scenarios or scenes (5%), and scenarios were misclassified as scenes in 14% of trials. Importantly, because the fMRI analyses aimed to capture the neural signatures of self-reported imagery experience, all analyses were repeated using participants' trial-by-trial classification responses rather than predefined stimulus categories. The fMRI classification task, in contrast, was used to link subjective imagery type to corresponding neural activation patterns. Although classification accuracy patterns were similar across both tasks, the fMRI analysis further incorporated individual classification responses to ensure that brain activity was modelled according to participants' experienced imagery type rather than the intended cue category.

**Vividness rating.** Vividness ratings showed a significant main effect of category, $F(3, 54) = 143.62$, $p < .001$, $\eta^2_p = .89$ (see Fig 2b). Post-hoc analyses confirmed that non-words were rated significantly lower than all imagery conditions ($p < .05$), while vividness did not differ between objects ($M = 2.98$, $SD = 0.71$), scenes ($M = 3.03$, $SD = 0.69$), and scenarios ($M = 3.18$, $SD = 0.74$). Non-words were rated lowest ($M = 1.31$, $SD = 0.54$). Because vividness did not differ across imagery types, it was not considered as a mediating variable in the multivariate analysis of brain activity.

**Reaction times.** Reaction times differed significantly across stimulus categories, $F(3, 54) = 17.09$, $p < .001$, $\eta^2_p = .49$. Post-hoc tests indicated that non-words were classified faster than all imagery conditions ($p < .05$), and that scenes were slower than both objects ($p = .004$) and scenarios ($p = .001$). Mean reaction times were 924.05 ms ($SD = 493$) for non-words, 1175ms ($SD = 615$) for objects, 1228 ms ($SD = 660$) for scenarios, and 1467ms ($SD = 795$) for scenes.

## Debriefing: Post-scan source memory task

Source memory was defined as recognition accuracy, reflecting participants' ability to correctly identify whether a stimulus was new or old and, if old, to correctly indicate the session in which it was originally presented (e.g., identifying a stimulus from the scanning session as belonging to that session). This varied across imagery categories and sessions (see Fig 3).

**Debriefing: Imagery category.** Recognition accuracy differed significantly across imagery categories, $F(2,36) = 6.70$, $p = .005$, $\eta^2 p = .35$. Post-hoc comparisons showed lower accuracy for scenarios compared to objects ($p = .003$). Mean recognition rates were 83% ($SD = 6\%$) for scenarios, 92% ($SD = 10\%$) for scenes, and 92% ($SD = 8\%$) for objects.

**Debriefing: Session.** Recognition accuracy differed significantly across sessions, $F(2, 36) = 60.02$, $p = < .001$, $\eta^2 p = .77$). Accuracy was lowest for stimuli from the eye-tracking session ($M = 38\%$, $SD = 27\%$), followed by stimuli from the scanning session ($M = 85\%$, $SD = 19\%$), and highest for novel stimuli ($M = 90\%$, $SD = 10\%$). Post-hoc comparisons confirmed significant differences between the eye-tracking stimuli and both the scanning and novel stimuli ($p < .001$). No significant interaction between imagery type and session was observed ($p = .15$).

When recognition accuracy was examined exclusively for stimuli from the scanning session, no category differences emerged ($p = .89$), suggesting that encoding or memory load during scanning did not differ between imagery categories and therefore cannot account for the observed fMRI effects.

**Debriefing question.** After scanning, all participants described significant differences between imagining objects, scenes and scenarios. Most participants reported that imagining objects began with a basic shape, which was then enriched with details – some perceived as 2D and others as 3D. Objects were typically imagined against a white background with some mentioning that the object slowly rotated in their mind's eye. In contrast, scenes were initially constructed as a 3D spatial framework, with individual elements and details integrated progressively to form a coherent, static image, resembling a photograph. A scenario, however, was described as a temporally extended, dynamically

**A**

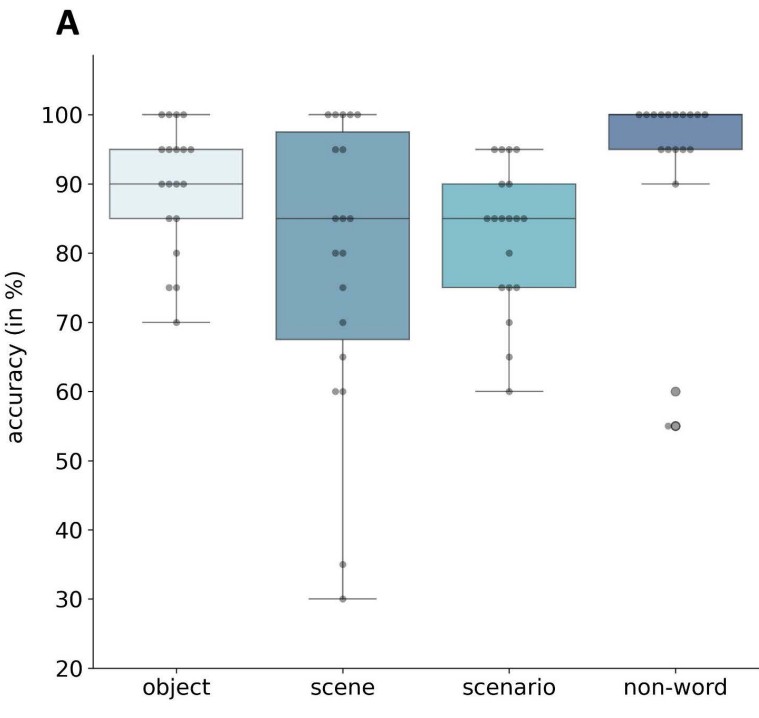

**B**

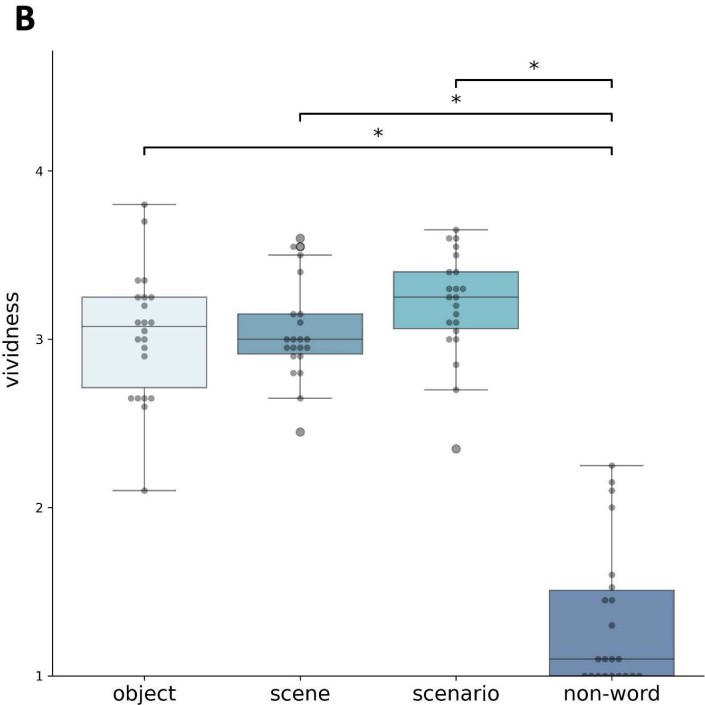

**Fig 2. Scanning task.** A: Classification accuracy (in percent) B: Vividness ratings. Higher scores resemble higher vividness ratings. *p < .05, **p < .01, ***p < .001.

**Fig 3. Post-scan Source Memory Task.** A: Bar graph showing the correct recognition (in percent) of stimuli as object, scene or scenario stimuli. B: Bar graphs showing the correct recognition (in percent) of stimuli from their respective session. *p < .05, **p < .01, ***p < .001.

unfolding sequence, characterized by movement of both people and oneself. Importantly, scenarios were always embedded within a spatially coherent context, which evolved over time, reinforcing the movie-like nature of these mental events. These descriptions support our view, that imagining objects, scenes or scenarios resemble qualitatively different cognitive experiences. Only two participants (which were excluded following the classification task during scanning) struggled with generating new content and reported relying on their memories in some of the trials.

### Task-based analysis of brain activity

The following paragraph includes the results of the task-dependent brain activation patterns and is followed by a task-dependent connectivity analysis.

**Visual Imagery Network: Mean-centered task-based PLS analysis of brain activation.** We used mean-centred task-based PLS analysis to assess neural differences between the four conditions. This analysis showed significant differences in brain patterns associated with imagery and the control condition ($p < .001$). Positive brain scores (or

saliences) were associated with regions in which the BOLD signal was greater for imagery. Negative brain scores (or saliences) were associated with regions in which the BOLD signal was greater for the control condition.

Within the imagery pattern, scenario construction had a stronger influence than scene and object construction, which made comparable contributions. This pattern encompassed bilateral vmPFC, anterior and posterior hippocampus, parahippocampal and lingual gyri, right precuneus, and bilateral middle occipital gyri. The control condition was associated with increased BOLD signal across a distributed fronto-parieto-occipital and lateral temporal brain pattern (see Fig 4 and S1 and S2 Tables).

**Scene and scenario construction network: Mean-centred task-based PLS analysis of brain activation.** Next, we repeated the task-based PLS analysis without the control condition to focus on differences between the imagery conditions. We found two significant LVs, representing (1) Scene construction and (2) Scenario construction.

**Scene construction.** The first LV ($p = .010$) differentiated scene (positive brains scores) from object construction (negative brain scores). Scenario construction did not contribute to this pattern (confidence intervals including zero, see Fig 5a). During scene construction, there was increased bilateral activity in the vmPFC, parahippocampal gyrus, lingual gyrus, and left lateral occipital lobe (see Fig 5b). In contrast, object construction was associated with fronto-parietal brain areas, cuneus, and posterior cingulate cortex.

**Scenario construction.** The second LV ($p = .026$) differentiated scenario (positive brain scores) from scene and object construction (negative brain scores, see Fig 5c). Scenario construction was supported by strong bilateral activation in the vmPFC, as well as in the left dorsolateral frontal areas, right anterior and posterior hippocampus, and bilateral posterior cingulate cortex, anterior precuneus, and angular gyrus. In contrast, scene/ object construction was associated with bilateral dorsal and lateral frontal cortex, bilateral posterior precuneus, bilateral fusiform gyri, right lingual gyrus, and inferior occipital lobe (see Fig 5d).

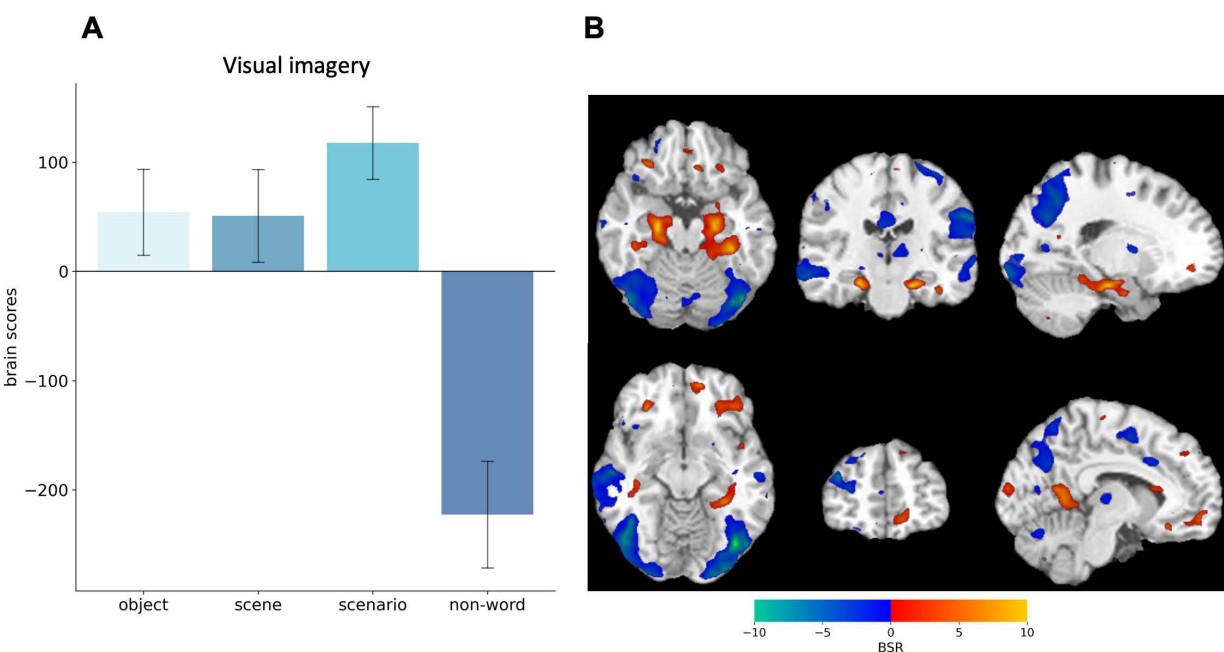

**Fig 4. Task-based PLS results from four conditions. (A)** Brain scores associated with visual imagery (LV1) differentiate object, scene, and scenario construction from the control task. Bar graphs display means with 95% bootstrapped confidence intervals. **(B)** Bootstrap Ratios (BSR) are displayed on a single-subject 1T template ni standard space. Warm colors indicate increased activity during imagery tasks, while cool colors represent increased activity during the control condition. The statistical map is thresholded at BSR = ±2.5.

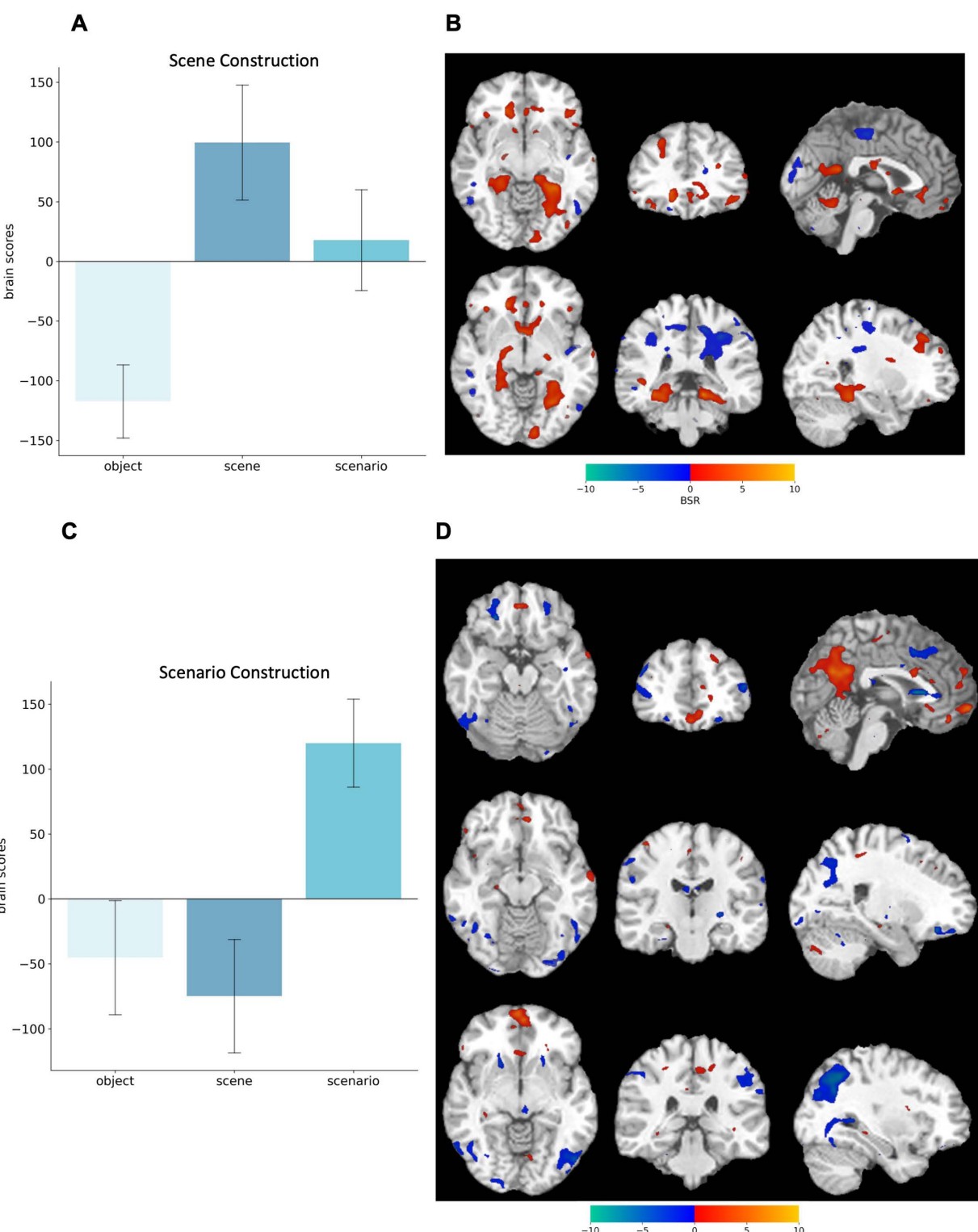

**Fig 5. Task-based PLS results from the imagery conditions. (A)** Brain scores associated with the first latent variable (LV1) differentiating scene from object construction, and **(C)** the second LV2 differentiating scenario from both object and scene construction. Bar graphs display means with 95% bootstrapped confidence intervals. **(B)** Bootstrap Ratios (BSR) are displayed on a single-subject 1T template ni standard space. Warm colours reflect activity

during scene construction, whereas cool colours reflect activity during object construction. **(D)** Warm colours reflect activity during scenario construction, whereas cool colours reflect activity during object and scene construction. The statistical map is thresholded at BSR=±2.5.

**Regional signal intensities.** We extracted signal intensities from ten a priori selected regions of interest (see Fig 6) to examine activity patterns across imagery conditions. Repeated-measures ANOVAs revealed condition-specific engagement patterns across several regions. In agreement with our primary hypothesis, the vmPFC showed significant modulation by condition ($F(2, 36) = 10.76$, $p < .001$, $\eta^2p = .37$), with stronger engagement during scenario construction compared to both scene ($p = .006$) and object construction ($p = .006$).

Medial temporal lobe regions also showed marginal to significant condition-specific effects. In the right posterior hippocampus ($F(2, 36) = 4.29$, $p = .021$, $\eta^2p = .19$) activation was stronger during scenario ($p = .061$) and scene ($p = .051$) compared with object construction, though these effects only approached significance. The right parahippocampal gyrus ($F(2, 36) = 9.53$, $p < .001$, $\eta^2p = .35$) showed greater activation during scene than object construction ($p = .003$) and marginally stronger activation during scenario construction ($p = .069$). Within the left parahippocampal gyrus ($F(2, 36) = 4.67$, $p = .01$, $\eta^2p = .21$) scenes elicited greater activation than objects ($p = .026$), with a marginal effect for scenarios ($p = .078$). We found no significant differences between conditions in the anterior hippocampi or in the posterior segment of the left hippocampus.

Within the precuneus, a gradation pattern was observed ($F(2, 36) = 5.62$, $p = .007$, $\eta^2p = .24$), in which scenario stimuli elicited stronger engagement than object stimuli ($p = .01$). In the visual perceptual cortex, hemispheric differences emerged. While no significant condition effects were observed in the right hemisphere, the left hemisphere showed condition-dependent activation ($F(2, 36) = 7.95$, $p = .001$, $\eta^2p = .30$), with stronger engagement during object ($p = .029$) and scene ($p = .002$) construction compared to scenario construction.

**Functional vmPFC-neocortical connectivity during visual imagery.** Since our main goal was to assess vmPFC contributions to imagery construction, we conducted a seed PLS analyses with the peak vmPFC voxel (MNI: −4 54–12) being the seed. Two significant LV's (see Fig 7 and S3 and S4 Tables) were identified, which showed a gradient in how the vmPFC functionally connects with the brain network.

The first LV ($p < .001$) showed a significant connectivity pattern during object ($r = −0.90$) and scene imagery ($r = −.83$), but not during scenario imagery ($r = −.14$, see Fig 7a). Notably, signal intensity changes in the vmPFC covaried with changes in bilateral fronto-parietal networks, anterior medial and posterior hippocampus, as well as precuneus and retrosplenial cortex (see Fig 7b).

The second LV ($p = .030$) differentiated scene and scenario construction from object construction (see Fig 7c). The strongest effect is observed in scenarios ($r = .89$), followed by scenes ($r = .69$), with objects showing an inverse relation ($r = −.33$). Regions exhibiting reliable functional connectivity during scenario and scene construction included bilateral anterior and posterior hippocampi, right parahippocampal cortex, and bilateral fusiform and lingual gyri. In contrast, object construction was associated with stronger functional connectivity with frontotemporal and frontoparietal networks (see Fig 7d).

## Discussion

This study utilized high-resolution 7T MRI to investigate the neural substrates supporting three forms of mental imagery. We proposed a graded hierarchical organization in which the posterior neocortex provides the visual-perceptual details, the hippocampus supports spatial scene construction, and the ventromedial prefrontal cortex (vmPFC) contributes to integrating these elements, with engagement increasing from single scenes to dynamic scenarios.

This study directly compares three forms of mental imagery: single objects without a spatial background, static isolated scenes, and naturalistically evolving mental scenarios. Since previous research has predominantly examined scene and object imagery in isolation, there is a critical gap in understanding the neural differences between these forms of mental

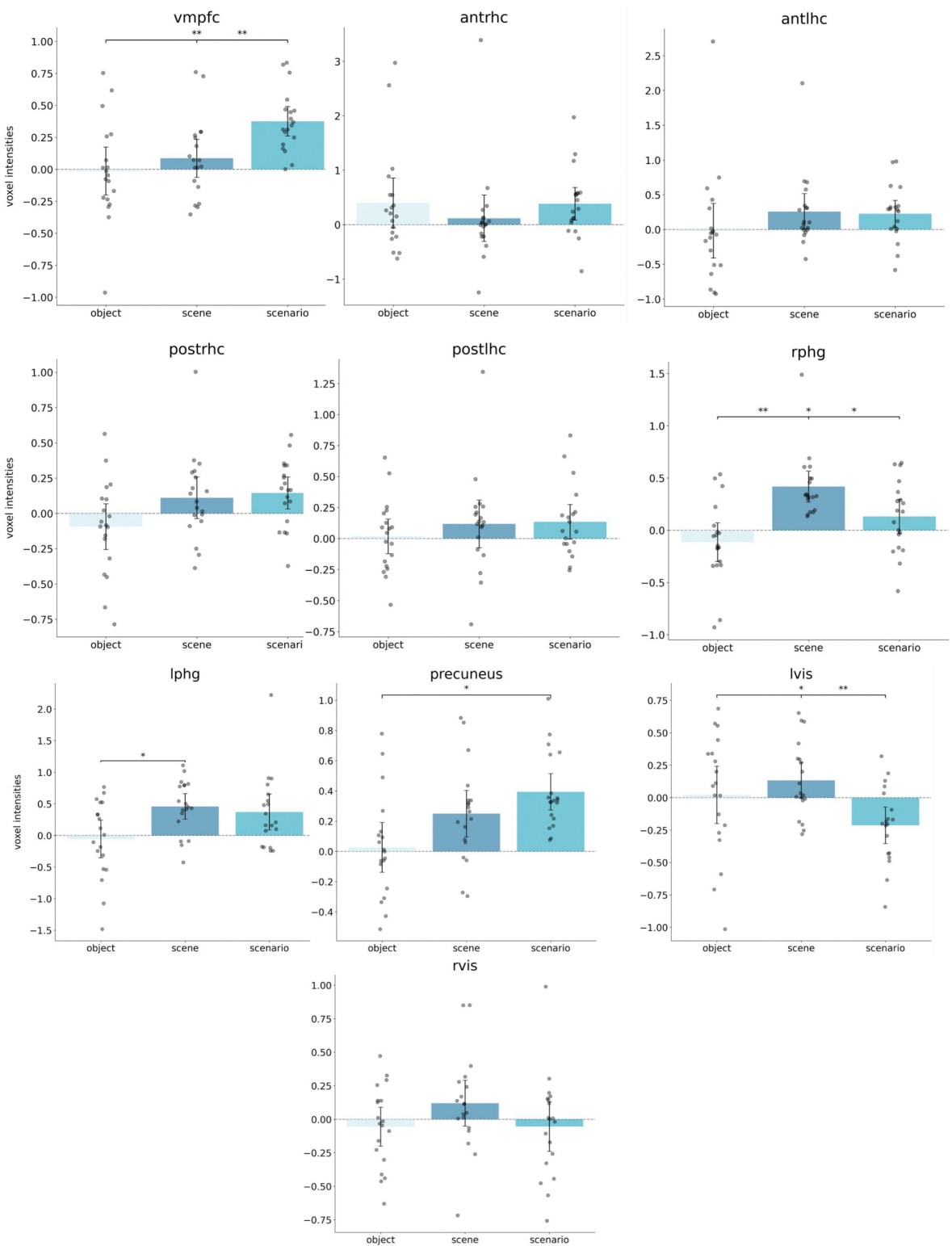

**Fig 6. Extracted signal intensities from brain regions associated with autobiographical memory taken from NeuroSynth.** Violin plots display the distribution of individual signal intensities. Box plots display the median and the interquartile range of voxel intensities. *p < .05, **p < .01, ***p < .001.

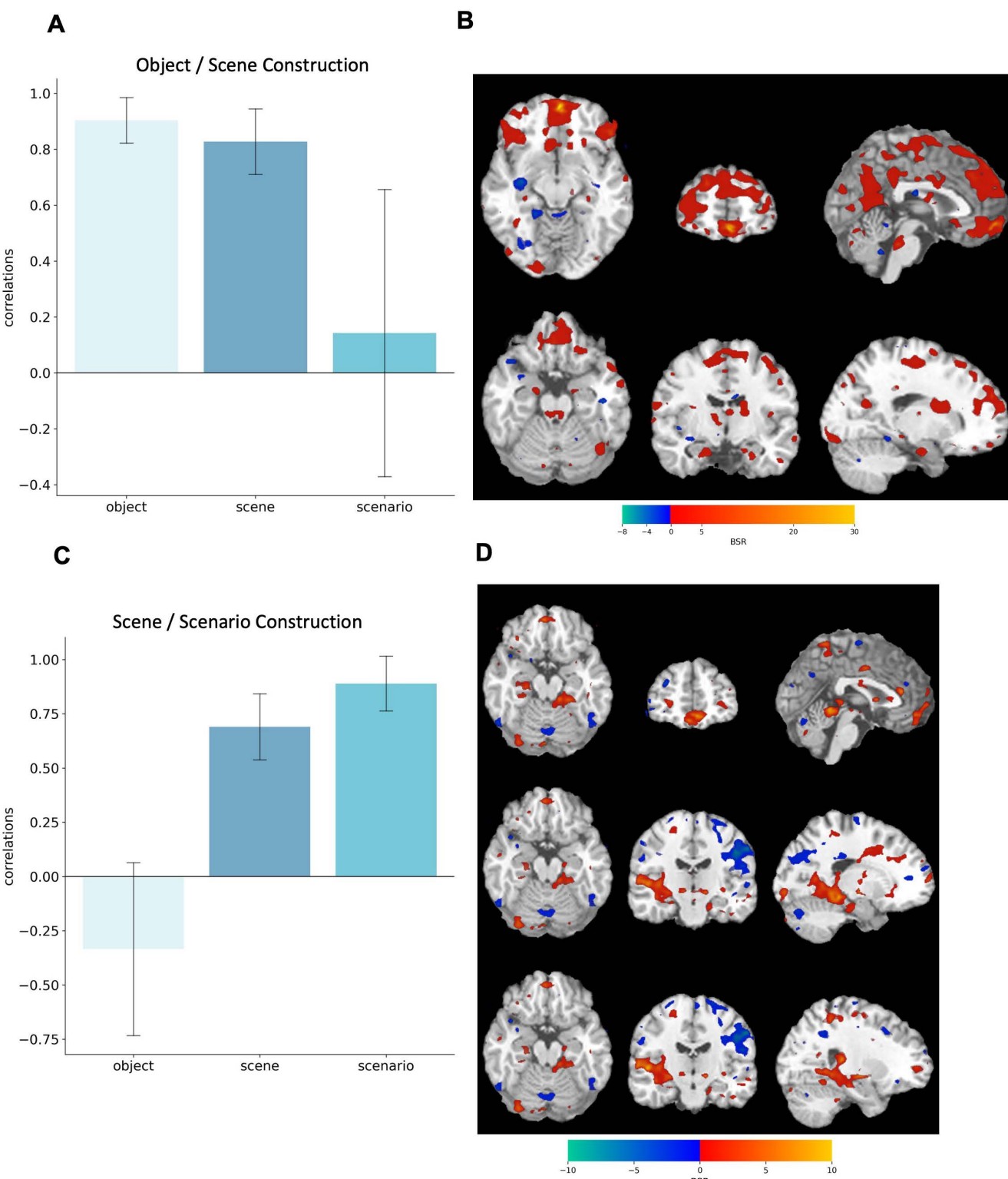

**Fig 7. Seed- PLS results.** Correlations associated with the significant latent variables LV1 **(A)** & LV2 **(C)** differentiating object, scene and scenario imagery. Bar graphs in (A) and (C) display means with 95% boot-strapped confidence intervals. **(B)** and **(D)** Bootstrap Ratios (BSR) are displayed on a single-subject T1 template in standard space. **(B)** Warm colours reflect connectivity during object and scene construction. **(D)** Warm colors reflect connectivity during scene and scenario construction. The statistical map is thresholded at BSR = ±2.0.

imagery [27,43–48]. We hypothesized that posterior cortices would support all imagery types by providing detailed visual information, that the hippocampus would contribute to constructing spatial contexts for scenes and scenarios, and that vmPFC involvement would increase with the temporal and contextual complexity of the imagined events.

### Common neural pattern for imagery-rich mental events

The activation pattern distinguishing imagery conditions from the low-level control task (letter counting in non-words) included the vmPFC, anterior/ posterior hippocampus, and the visual-perceptual cortex, a network consistently implicated in mental imagery across numerous studies [24–26,49–53]. This convergence supports the idea that object, scene, and scenario construction share a common imagery network that varies in degree of engagement across regions.

### vmPFC: Temporal integration and scenario construction

Our findings support a hierarchical account in which the vmPFC contributes to both scene and scenario construction, but with differential degrees of engagement depending on the complexity and temporal structure of the mental representation. The vmPFC appears to play a modulatory role that becomes progressively stronger as mental events extend over time and integrate multiple spatial contexts.

This was demonstrated by multiple analyses: (1) In the mean-centred PLS (including all conditions), scenario construction showed the strongest association to a neural pattern that included the vmPFC, anterior/ posterior hippocampus, and visual-perceptual cortices – regions consistently linked to scene processing. (2) In the mean-centred PLS (imagery conditions only), we identified a significant latent variable that distinguished scenario from scene construction, showing that scenario construction was associated with a distinct neural pattern that prominently involved the vmPFC. (3) A priori selected ROI analyses revealed that the vmPFC was more strongly engaged during scenario than both scene and object construction. (4) Functional connectivity analysis with the vmPFC as a seed revealed that during scenario and scene construction, the vmPFC exhibited stronger positive connectivity with a widespread brain network, including the hippocampus and visual-perceptual cortices, compared to object construction.

Together, these results suggest that vmPFC is not exclusively involved in scenario construction but contributes to integrative processing across imagery types, supporting the synthesis of spatial, contextual, and semantic features into coherent mental representations. In line with our findings, neuropsychological and neuroimaging evidence converge in showing that both the hippocampus and vmPFC contribute to constructing coherent scenes and scenarios. Lesion studies have demonstrated that vmPFC patients, similar to hippocampal patients, show impairments in scene construction tasks, such as boundary extension, episodic future thinking and autobiographical memory tasks [7,15,16,54–56]. However, these deficits differ qualitatively: vmPFC damage appears to disrupt the organization or coherence of constructed scenes, whereas hippocampal damage affects the generation of spatial detail and structure.

Furthermore, recent MEG studies have provided evidence that the vmPFC exerts a driving influence on the hippocampus during tasks that require the temporal unfolding of mental events [20–22]. The present results support this view, showing a continuous increase in vmPFC engagement from objects to scenes to scenarios, consistent with a hierarchical and interactive model of imagery construction.

Our data further suggest a qualitative difference in the spatial engagement of the vmPFC during scenario versus scene construction. Scenarios appeared to recruit vmPFC regions that extended more anteriorly, and both dorsally and ventrally compared to scene imagery. This spatial pattern aligns with recent evidence indicating a graded functional organization within the vmPFC, characterized by distinct subfields and connectivity profiles [57]. Scenarios, which require temporal unfolding and integration of multiple scenes, may therefore preferentially engage more anterior and transmodal subregions of the vmPFC, whereas single-scene construction elicits relatively posterior and spatially circumscribed vmPFC activation. However, interpretation of these results is limited by the study design, which precluded formal testing of the underlying hypotheses. These issues should be addressed in future studies.

## Hippocampus: Scene construction

Our findings indicate that the hippocampus and surrounding medial temporal lobe structures support scenario and scene construction to a greater extent than object construction. The mean-centred PLS revealed two significant LVs: LV1 showed greater hippocampal/parahippocampal activation for scenes vs. objects, while LV2 showed greater hippocampal activation for scenarios vs. objects. ROI analyses confirmed this pattern, particularly in the right parahippocampal gyrus and the left parahippocampal gyrus.

Previous research connects scene construction to the hippocampus [3–5,7,12,28], its anterior segment [24–27]. However, the pattern was not corroborated by the ROI analysis, which failed to reveal a strong preference of the anterior hippocampus for scene versus object construction. This discrepancy may be attributed to our approach of preselecting ROIs based on the Neurosynth database. While this approach provides a data-driven method for ROI selection based on large-scale neuroimaging findings, it does carry the inherent risk of missing activation peaks that may be specific to our sample and experimental paradigm. The coordinates derived from meta-analyses represent population-level averages and may not align with the precise anatomical or functional peaks present in individual studies, particularly when examining specific cognitive processes or using novel experimental paradigms. This limitation is especially relevant for regions like the anterior hippocampus, where subtle spatial differences in activation peaks can have significant functional implications. Future investigations would benefit from complementary approaches, such as subject-specific functional localizers or data-driven methods that can identify sample-specific activation patterns, to ensure comprehensive capture of relevant neural activity. It is worth noting that a subsequent analysis of this dataset, which extends beyond the scope of the current investigation, will examine hippocampal subfields in greater detail to elucidate their specific contributions to the construction of mental imagery-rich events.

## Posterior neocortex: Sensory-perceptual support

The posterior neocortex was engaged in all imagery conditions, as shown in the first PLS analysis contrasting all imagery conditions to the low-level baseline. The areas comprised the retrosplenial cortex, precuneus, lingual, and fusiform gyrus, as well as middle occipital gyrus. We hypothesised previously that the posterior neocortex provides visual-perceptual elements to any type of mental imagery-rich events, such as colour, shape and movement details [58–63]. Since object construction necessitates focussed attention on such details, it is fitting that the ROI analysis of the left visual cortex revealed greater activation during object construction relative to other types of mental imagery.

Higher engagement of the precuneus during scenario construction, as indicated by the PLS and ROI analyses is likely due to its role in self-referential processing, temporal and spatial integration of complex information, which is essential for mentally constructing spatiotemporally coherent mental simulations [64,65].

## Behavioural and eye movement differences in response to cognitive demands during visual imagery

Lastly, the behavioural and oculomotor data provide further critical insights into the constituents of scenario construction distinguishing it from scene and object construction. The imagery conditions were carefully designed and piloted to ensure that vividness, construction demands, and word length were consistent across conditions, isolating the precise mental content as the key difference.

During object construction, participants reported focusing on a single, detailed object devoid of spatial context, confirmed by less frequent smaller-amplitude saccades and long fixations compared to scenes. In contrast, scene construction was described to involve immediately generating a spatial layout, subsequently filled with multiple visuospatial details, reflected in frequent, short fixations, and large-amplitude saccades – indicative of constructing a more spatially extended mental space. This pattern aligns with prior findings on scene construction [4,30,41,42].

Analysis of reaction time data during the classification tasks, it was consistently shown that scenarios were classified faster than scenes, while objects were intermediate. These findings indicate that temporally extended, event-like

scenarios are generated and classified more fluently than individual scenes, whereas objects and scenes show more comparable processing. These results are in line with the oculomotor data.

While temporally extended scenarios were remembered and visualized as vividly and accurately as scenes and objects, participants described scenario construction as a sequential and combinatory process, starting with rapid spatial layout generation (like scenes) and followed by the construction of individual foreground objects integrated into an unfolding narrative. Interestingly, despite the similarity in neuronal activation patterns between scenarios and scenes, eye-tracking data revealed that scenarios and objects triggered more similar fixation durations and saccade amplitudes than scenes. This apparent discrepancy may be explained by the dual processing demands of scenario construction: object-focused visual attention, as captured by eye-tracking metrics [30,66,67], and scene-based integration, as reflected in fMRI data. Together, these findings highlight the complex interplay between visual attention and neural mechanisms in constructing temporally extended mental events, underscoring the unique cognitive demands of scenario construction.

## Limitations

We acknowledge that our sample size of 19 participants may limit the generalizability of our findings. However, several methodological factors help mitigate these limitations. First, our use of ultra-high field 7T fMRI provides enhanced spatial resolution and signal-to-noise ratio compared to standard field strengths, potentially increasing statistical power for detecting neural activation patterns. Second, our multivariate PLS approach is particularly well-suited for smaller samples as it reduces dimensionality while capturing covariance patterns across the entire brain, making it more sensitive to distributed neural networks than traditional univariate approaches. Third, our within-subjects design with multiple conditions per participant increases statistical power by reducing between-subject variance. Replication in larger samples will be essential to establish the robustness and broader applicability of the neural mechanisms we have identified for scenario construction.

Second, by instructing participants to construct novel objects, scenes or scenarios, we wanted to minimize inter-individual variability related to personal memories and to isolate the cognitive and neural mechanisms underlying scene and scenario construction, independent of autobiographical content. The precise degree to which autobiographical recall contributed to participants' mental simulations during the eye-tracking or fMRI experiment cannot be determined. Thus, while our design minimized explicit autobiographical retrieval, the findings should be interpreted considering the overlap between memory-based and constructive forms of imagery.

## Conclusion

Our study provides a comprehensive neural framework for how the brain constructs naturalistic, imagery-rich events. The posterior neocortex provides the perceptual details, the hippocampus supports the spatial layout essential for scene and scenario construction, and the vmPFC supports integrative and temporal organization processes that scale with representational complexity. These findings refine current models of mental imagery and offer a foundation for investigating its impairments in clinical conditions further.

## Supporting information

**S1 Table. Cluster report for the first latent variable from the mean-centered Task-based PLS with the imagery conditions.**
(DOCX)

**S2 Table. Cluster report for the second latent variable from the mean-centered Task-based PLS with the imagery conditions.**
(DOCX)

**S3 Table. Cluster report from the first latent variable from the mean centered task-based Seed PLS.**
(DOCX)

**S4 Table. Cluster report from the second latent variable from the mean centered task-based Seed PLS.**
(DOCX)

## Acknowledgments

The authors would like to thank A. Ruehling for excellent technical assistance during the acquisition of the imaging data. The imaging experiments were performed at the German Center for Neurodegenerative Diseases (DZNE), Bonn, Germany.

## Author contributions

**Conceptualization:** Cornelia McCormick.

**Data curation:** Julia Taube, Pitshaporn Leelaarporn, Rüdiger Stirnberg, Yilmaz Sagik.

**Formal analysis:** Julia Taube, Pitshaporn Leelaarporn, Maren Bilzer, Rüdiger Stirnberg, Yilmaz Sagik, Cornelia McCormick.

**Funding acquisition:** Cornelia McCormick.

**Investigation:** Julia Taube, Pitshaporn Leelaarporn, Cornelia McCormick.

**Methodology:** Julia Taube, Pitshaporn Leelaarporn, Rüdiger Stirnberg, Cornelia McCormick.

**Project administration:** Cornelia McCormick.

**Resources:** Cornelia McCormick.

**Supervision:** Julia Taube, Pitshaporn Leelaarporn, Cornelia McCormick.

**Validation:** Julia Taube, Pitshaporn Leelaarporn, Maren Bilzer, Yilmaz Sagik, Cornelia McCormick.

**Visualization:** Julia Taube.

**Writing – original draft:** Julia Taube, Pitshaporn Leelaarporn, Cornelia McCormick.

**Writing – review & editing:** Julia Taube, Pitshaporn Leelaarporn, Maren Bilzer, Rüdiger Stirnberg, Yilmaz Sagik, Cornelia McCormick.

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
