## [Decision Letter · Decision Letter 0]

24 Aug 2025

Dear Dr. McCormick,

Thank you for submitting your manuscript to PLOS ONE. After careful consideration, we feel that it has merit but does not fully meet PLOS ONE’s publication criteria as it currently stands. Therefore, we invite you to submit a revised version of the manuscript that addresses the points raised during the review process.

We look forward to receiving your revised manuscript.

Kind regards,

Akitoshi Ogawa, Ph.D.

Academic Editor

PLOS ONE

Journal Requirements:

This research was supported by the Hertie Network of Excellence in Clinical Neuroscience. Work in C.M.’s lab is further financed by internal research funding of the Fac-ulty of Medicine (BONFOR), University Hospital Bonn, by the Federal Ministry of Edu-cation and Research (BMBF) within the framework of the funding programme ACCENT (funding code 01EO2107) and by the Deutsche Forschungsgemeinschaft (DFG, German Research Foundation, MC 244/3-1).  J.T. received an Argelander Mobility Grant from the University Bonn.

4. In this instance it seems there may be acceptable restrictions in place that prevent the public sharing of your minimal data. However, in line with our goal of ensuring long-term data availability to all interested researchers, PLOS’ Data Policy states that authors cannot be the sole named individuals responsible for ensuring data access (http://journals.plos.org/plosone/s/data-availability#loc-acceptable-data-sharing-methods).

Reviewers' comments:

Reviewer's Responses to Questions

**Comments to the Author**

1. Is the manuscript technically sound, and do the data support the conclusions?

Reviewer #1: Yes

Reviewer #2: Yes

2. Has the statistical analysis been performed appropriately and rigorously?

Reviewer #1: Yes

Reviewer #2: Yes

3. Have the authors made all data underlying the findings in their manuscript fully available?

Reviewer #1: Yes

Reviewer #2: Yes

4. Is the manuscript presented in an intelligible fashion and written in standard English?

Reviewer #1: Yes

Reviewer #2: Yes

Reviewer #1: Using ultra-high field fMRI, the authors investigated (differences in) the neural bases of the construction of objects, scenes, and scenarios. They report that a shared neural network, including vmPFC, the hippocampus, and the posterior neocortex, was engaged in all types of mental imagery. The hippocampus was mainly involved in the construction of scenes and scenarios, while the vmPFC was maximally involved in the construction of scenarios.

I think this is an interesting study, carefully conducted and using a nice experimental paradigm. I enjoyed reading it throughout. I also appreciated the reporting of participants’ personal strategies to approach different imagery tasks.

I have a few suggestions to improve the completeness and clarity of the paper.

There is a strong emphasis in the paper (both in the Introduction and the Discussion) on the involvement of vmPFC in the construction of scenarios, and at times it is stated that vmPFC is not involved in the construction of single scenes. The literature cited in support of this claim provides some indirect evidence (to me, not even convincing) that this is the case. On the other hand, the most stringent neuropsychological research on the involvement of vmPFC in the construction of single scenes, testing hippocampal and vmPFC patients in boundary extension, showed that vmPFC patients - like hippocampal patients - were impaired in scene construction. In that same paper, the authors found however differences between the scene construction impairments of hippocampal and vmPFC patients. In fact, even the results of this experiment do not run against the idea that vmPFC is implicated in scene construction, even though (perhaps) less than it is implicated in scenario construction.

This paper should be cited and discussed: De Luca F, McCormick C, Mullally SL, Intraub H, Maguire EA, Ciaramelli E. (2018). Boundary extension is attenuated in patients with ventromedial prefrontal cortex damage. Cortex, 108, 1-12.

Another inappropriate citing of the literature is on p. 3, line 65. Mind-wandering is reduced in vmPFC patients BUT NOT in patients with hippocampal lesions, in whom mind-wandering is as frequent as in controls but less episodic in nature. Please correct.

At the end of the Introduction, the author should sketch the study design and the predictions, including the eye movement experiment and its rationale.

Was any participant in the depression range/excluded for their score at the Depression scale? P. 6.

The authors say that the stimuli were closely matched words. Given that these were not single words, maybe it would be more appropriate to call them in a different way, like verbal cues, verbal expressions, verbal labels. Also, these verbal cues were matched on what? Please specify.

It is stated that an independent sample of participants rated the type of imagery triggered by these “words”. Which were the precise instructions given to these participants for the rating? (It is never clear to me in reading the paper how participants get to know that they have to imagine things differently depending on imagery category, see below).

p. 7, line 157: based on accuracy you mean classification accuracy?

p. 8 line 154: they completed a practice session: practice of what task? Imagery? Classification? Both?

In the fMRI task, upon seeing the verbal cue for imagination (e.g., a busy cafè), did participants also receive the instruction that they had to imagine it as a scene (as opposed for example to a scenario?). I understand that they were instructed on different types of imagery before the task, but did they receive an instruction to imagine something as object/scene/scenarios upon seeing the verbal cue? Please clarify in the paper, or maybe show an experimental trial. Same applies to the eye tracking experiment. Again on p. 11, it is stated that in fMRI participants were instructed to visualise the stimuli. With which instructions? Just visualization with no instruction (passive viewing)?

I do not think that reaction times in the classification task are a relevant measure, or maybe I do not clearly find what was made of this variable. I had to go back and forth several times in the paper to understand whether there also was an analysis of response times for imagery Not clear.

I do not understand the difference in the results reported for the eye tracking task (p. 16, line 363) and for the scanning task (p. 17, line 405). What did I misunderstand? Please clarify and improve the readability of the section.

Did imagining a scenario take longer than imagining a scene or an object? Couldn’t the different findings on the neural bases of imagining objects, scene, scenarios just reflect time on task?

There is not much/not at all discussion of the findings from the eye movement experiment or of the source memory experiment. Not much was made of these data.

From the PLS results I see that vmPFC was implicated in both scene construction and, perhaps more, scenario construction. Were vmPFC regions implicated in scene and scenario construction differ in the spatial coordinates? Might there be subregions of vmPFC subserving the imagination of single scenes and the temporal unfolding of scenes in a scenario?

On p. 27, line 632, again a wrong citation: Bertossi & Ciaramelli, 2016 is a study about mind-wandering, not autobiographical memory and future thinking.

Maybe you referred to other studies, for example:

Bertossi E, Tesini C, Cappelli A, Ciaramelli E (2016). Ventromedial prefrontal damage causes a pervasive impairment of episodic memory and future thinking. Neuropsychologia, 90, 12-24.

Ciaramelli, E., Anelli, F., & Frassinetti, F. (2021). An asymmetry between past and future mental time travel following vmPFC damage. Social Cognitive and Affective Neuroscience, 16, 315-25.

Reviewer #2: Overall Evaluation

The work is well-structured, clear in its theoretical framework, and methodologically solid. The use of 7T MRI, combined with carefully controlled imagery paradigms, represents an original and significant contribution to the study of the neural processes underlying the construction of mental events. The multivariate PLS analysis and the combination of behavioral, oculomotor, and fMRI data provide a robust triangulation of the conclusions.

However, there are several aspects that could be improved to strengthen the clarity, replicability, and impact of the work.

Major points

“Scenario” is defined as a temporally extended mental event, but at times it seems to overlap with “autobiographical episode” in the discussion. It may be helpful to clarify explicitly to what extent the scenarios are or are not autobiographical. Why was it important that the scenarios constructed by participants were not autobiographical episodes? From which perspective were the scenes and scenarios imagined (first or third person)?

The manuscript states: “Extensive training was given to ensure that participants understood these instructions correctly and were able to differentiate between the three imagery conditions.” It would be useful to describe in more detail how this training was conducted.

Sample size: N = 19 is relatively small for an fMRI study, although power may be increased by the 7T resolution and PLS analysis. It would be useful to discuss the limits of generalizability and possible interindividual variability.

Were participants excluded only in the “Eye-tracking data analysis”? The text reports that 22 data points were removed; it would be helpful to specify how many participants in total were excluded. In repeated-measures ANOVA, if a data point is removed, the entire participant must typically be excluded.

Some effects known in the literature, such as preference for scenes over individual objects in the anterior hippocampus, were not found. This may be due to the sample size.

Predefined ROIs: ROIs were selected using Neurosynth. While this is a valid approach, it would be worth discussing the risk of missing activation peaks specific to the sample (as already mentioned for the anterior hippocampus) and the implications of this.

Control for autobiographical memory: Despite instructions to avoid personal memories, two participants reported relying on autobiographical recall. It may be useful to analyze the data with and without these participants to verify the robustness of the results.

It should also be clarified how the researchers determined which participants had difficulty constructing non-autobiographical scenarios.

Connectivity: Positive and negative connectivity in the PLS is interpreted in functional terms, but it would be useful to specify whether such patterns have been observed in similar studies and to discuss alternative interpretations.

It would also be interesting to test whether vividness correlates with the fMRI signal, regardless of the lack of differences between categories.

Finally, please justify why Bonferroni correction was used in some cases and Holm–Bonferroni in others, and clarify the criteria for this choice.

Results: this section is not immediately easy to follow. It would be preferable to restructure it by first reporting the significant factors and then the post hoc tests with their corresponding means.

When reporting differences between scenes and both scenarios and objects, please also indicate whether scenarios and objects statistically differ from each other.

Minor points:

“ni standard space”

In the Introduction (Line 63, page 3), a period is missing at the end of a sentence.

The authors may consider adding a paragraph on clinical implications (e.g., in patients with vmPFC or parietal damage) and a sensitivity analysis excluding the two participants who relied on autobiographical memory.

**Do you want your identity to be public for this peer review?** For information about this choice, including consent withdrawal, please see our Privacy Policy

Reviewer #1: No

Reviewer #2: No

---

## [Author Response · Author response to Decision Letter 1]

22 Oct 2025

Dear Reviewers,

we thank the reviewers for their thoughtful and constructive feedback, which has greatly helped us to clarify and strengthen the manuscript. We have carefully con-sidered each comment and believe that the revisions have substantially improved the clarity, coherence, and interpretive depth of the manuscript.

Below, we provide a point-by-point response to each of the reviewer’s comments. Reviewer comments are reproduced in black, followed by our detailed responses in blue. All major textual changes have been incorporated into the revised version and are highlighted accordingly.

We hope that the revised version meets the reviewers’ expectations and provides a clearer account of our findings and their contribution to the literature on mental im-agery and hippocampal–prefrontal interactions.

With kind regards

Cornelia McCormick

(on behalf of all co-authors)

Reviewer #1

Using ultra-high field fMRI, the authors investigated (differences in) the neural ba-ses of the construction of objects, scenes, and scenarios. They report that a shared neural network, including vmPFC, the hippocampus, and the posterior neocortex, was engaged in all types of mental imagery. The hippocampus was mainly involved in the construction of scenes and scenarios, while the vmPFC was maximally in-volved in the construction of scenarios. I think this is an interesting study, carefully conducted and using a nice experimental paradigm. I enjoyed reading it throughout. I also appreciated the reporting of participants’ personal strategies to approach dif-ferent imagery tasks.

I have a few suggestions to improve the completeness and clarity of the paper.

There is a strong emphasis in the paper (both in the Introduction and the Discus-sion) on the involvement of vmPFC in the construction of scenarios, and at times it is stated that vmPFC is not involved in the construction of single scenes. The literature cited in support of this claim provides some indirect evidence (to me, not even con-vincing) that this is the case. On the other hand, the most stringent neuropsycholog-ical research on the involvement of vmPFC in the construction of single scenes, test-ing hippocampal and vmPFC patients in boundary extension, showed that vmPFC patients - like hippocampal patients - were impaired in scene construction. In that same paper, the authors found however differences between the scene construction impairments of hippocampal and vmPFC patients. In fact, even the results of this experiment do not run against the idea that vmPFC is implicated in scene construc-tion, even though (perhaps) less than it is implicated in scenario construction. This paper should be cited and discussed: De Luca F, McCormick C, Mullally SL, Intraub H, Maguire EA, Ciaramelli E. (2018). Boundary extension is attenuated in patients with ventromedial prefrontal cortex damage. Cortex, 108, 1-12.

Response: Our emphasis of the vmPFC as primarily involved in scenario construc-tion while minimally involved in single scene construction may have been overstat-ed. The reviewer's point about the boundary extension research providing evidence for vmPFC contributions to scene construction has been added to the manuscript.

In addition, we would like to clarify our position: rather than arguing that vmPFC is not involved in single scene construction, our emphasis was on the relative contri-bution of the vmPFC across these processes. The neuropsychological evidence the reviewer cites actually supports this nuanced view. While both hippocampal and vmPFC patients showed scene construction impairments, the differences between their impairment patterns suggest distinct but complementary roles for these re-gions. We added this information into the manuscript and reframed our argument in terms of relative contributions rather than presence/absence. We’ve accordingly adapted our abstract (p.2, l. 38-43), introduction (p.3, l.66 – p.4, l. 117) and the dis-cussion (p.27, l.902 – p.30, l.1005).

Another inappropriate citing of the literature is on p. 3, line 65. Mind-wandering is reduced in vmPFC patients BUT NOT in patients with hippocampal lesions, in whom mind-wandering is as frequent as in controls but less episodic in nature. Please cor-rect.

Response: We thank the reviewer for pointing out the inappropriate citing. This mis-take has now been corrected in the manuscript.

New paragraph:

First, previous research described that the spontaneous initiation of internal mental events (e.g., mind-wandering episodes) appeared not only reduced in vmPFC-damaged patients (Bertossi & Ciaramelli, 2016), but also their off-task thoughts re-lated less to the future and more to the present in comparison to controls. Important-ly, the presence of hippocampal lesions did not lead to a measurable decrease of mind-wandering episodes, but their episodes appeared less visual and verbal se-mantic in nature (Costi et al., 2018; Luelsberg et al., 2022; McCormick, Rosenthal, et al., 2018).

At the end of the Introduction, the author should sketch the study design and the predictions, including the eye movement experiment and its rationale.

Response: We thank the reviewer for pointing out that we missed a short study de-sign description and the missing rationale for the eye-tracking experiment. We have now included this information as a new paragraph. See p.6, l. 171-185

Was any participant in the depression range/excluded for their score at the Depres-sion scale? P. 6.

Response: We have clarified in the manuscript that no participants were excluded based on their BDI-V score.

New paragraph: No participants were excluded based on their BDI-V score.

The authors say that the stimuli were closely matched words. Given that these were not single words, maybe it would be more appropriate to call them in a different way, like verbal cues, verbal expressions, verbal labels.

Response: We agree with the reviewer’s suggestion and now consistently refer to the stimuli as verbal cues throughout the manuscript.

Also, these verbal cues were matched on what? Please specify.

Response: We apologize for the lack of clarity. The verbal cues were matched on word length (see p. 8, l. 249-250). Additionally, post hoc analyses indicated that cue vividness did not differ between imagery conditions (see p. 18, l. 529 & p.20, l. 624-627).

It is stated that an independent sample of participants rated the type of imagery trig-gered by these “words”. Which were the precise instructions given to these partici-pants for the rating? (It is never clear to me in reading the paper how participants get to know that they have to imagine things differently depending on imagery cate-gory, see below).

Response: We apologize for the lack of clarity. We now included two paragraphs explaining the instructions in more detail for the selection of stimuli (p. 7, l. 215 – 236) and during eye-tracking/fmri (p. 9, l. 275-313).

p. 7, line 157: based on accuracy you mean classification accuracy?

Response: Yes, we included this detail into the new paragraph.

New Paragraph: Based on these ratings, we selected cues using a discrimination-accuracy criterion: only cues that consistently elicited a single imagery category above a prespecified threshold were retained.

p. 8 line 154: they completed a practice session: practice of what task? Imagery? Classification? Both?

Response: We’ve added a new paragraph to clarify the course of the practice ses-sion. See p. 9, l. 262-273

In the fMRI task, upon seeing the verbal cue for imagination (e.g., a busy cafè), did participants also receive the instruction that they had to imagine it as a scene (as opposed for example to a scenario?). I understand that they were instructed on dif-ferent types of imagery before the task, but did they receive an instruction to imagine something as object/scene/scenarios upon seeing the verbal cue? Please clarify in the paper, or maybe show an experimental trial. Same applies to the eye tracking experiment. Again on p. 11, it is stated that in fMRI participants were instructed to visualise the stimuli. With which instructions? Just visualization with no instruction (passive viewing)?

Response: We’ve clarified the fmri / eye-tracking task in more detail in a new para-graph. See p. 9, l. 275-313

I do not think that reaction times in the classification task are a relevant measure, or maybe I do not clearly find what was made of this variable. I had to go back and forth several times in the paper to understand whether there also was an analysis of response times for imagery Not clear.

Response: We thank the reviewer for this helpful comment and agree that our initial description of the reaction time (RT) measure required clarification. We have now added a paragraph clarifying the rationale for including reaction times in the classification task (p.14, l. 420-423), a new paragraph in the Results section (p.18, l.533-549) and the Discussion (p.32, l. 1062-1066). We deleted the reaction times for the vividness rating.

I do not understand the difference in the results reported for the eye tracking task (p. 16, line 363) and for the scanning task (p. 17, line 405). What did I misunderstand? Please clarify and improve the readability of the section.

Response: We have changed the paragraph to improve readability. Please see p.18, l.520-527 for eye tracking task results and p.20, l. 607-622 for scanning task.

Did imagining a scenario take longer than imagining a scene or an object? Couldn’t the different findings on the neural bases of imagining objects, scene, scenarios just reflect time on task?

Response: We added a paragraph to highlight that there were no differences in vis-ualization duration between the imagery conditions.

New paragraph: Each stimulus was displayed for 10 seconds, and participants were instructed to engage in the respective imagery task throughout the entire presenta-tion period.

There is not much/not at all discussion of the findings from the eye movement exper-iment or of the source memory experiment. Not much was made of these data.

Response: We agree that the discussion of the eye-tracking and source memory findings was relatively brief. These experiments were primarily included as support-ing measures to confirm task engagement and stimulus differentiation, rather than as central tests of our hypotheses. Therefore, the results were reported for com-pleteness but not discussed in depth, as our main focus lay on the neural mecha-nisms underlying imagery generation during fMRI.

From the PLS results I see that vmPFC was implicated in both scene construction and, perhaps more, scenario construction. Were vmPFC regions implicated in sce-ne and scenario construction differ in the spatial coordinates? Might there be sub-regions of vmPFC subserving the imagination of single scenes and the temporal unfolding of scenes in a scenario?

Response: We thank the reviewer for this insightful comment. We added a short paragraph discussing possible spatial differences on p. 29, l. 980-1006.

On p. 27, line 632, again a wrong citation: Bertossi & Ciaramelli, 2016 is a study about mind-wandering, not autobiographical memory and future thinking.

Maybe you referred to other studies, for example:

Bertossi E, Tesini C, Cappelli A, Ciaramelli E (2016). Ventromedial prefrontal dam-age causes a pervasive impairment of episodic memory and future thinking. Neuro-psychologia, 90, 12-24.

Ciaramelli, E., Anelli, F., & Frassinetti, F. (2021). An asymmetry between past and future mental time travel following vmPFC damage. Social Cognitive and Affective Neuroscience, 16, 315-25.

Response: We thank the reviewers for this helpful remark. The citation has been corrected, and the suggested studies have been added accordingly.

Reviewer #2:

Overall Evaluation

The work is well-structured, clear in its theoretical framework, and methodologically solid. The use of 7T MRI, combined with carefully controlled imagery paradigms, represents an original and significant contribution to the study of the neural pro-cesses underlying the construction of mental events. The multivariate PLS analysis and the combination of behavioral, oculomotor, and fMRI data provide a robust tri-angulation of the conclusions.

However, there are several aspects that could be improved to strengthen the clarity, replicability, and impact of the work.

Major points

“Scenario” is defined as a temporally extended mental event, but at times it seems to overlap with “autobiographical episode” in the discussion. It may be helpful to clarify explicitly to what extent the scenarios are or are not autobiographical. Why was it important that the scenarios constructed by participants were not autobio-graphical episodes? From which perspective were the scenes and scenarios imag-ined (first or third person)?

Response: We appreciate the reviewer’s thoughtful comment regarding the poten-tial overlap between scenario imagery and autobiographical episodic recall. We acknowledge that mental imagery and autobiographical memory inherently share cognitive and neural processes, such as scene construction. Thus, while partici-pants were instructed to imagine novel scenarios rather than recall specific past events, we cannot exclude the possibility that some imagery trials involved autobio-graphical elements. Therefore, the precise degree to which autobiographical recall contributed to participants’ mental simulations cannot be determined. However, by instructing participants to construct novel objects, scenes or events, we wanted to minimize inter-individual variability related to personal memories and to isolate the cognitive and neural mechanisms underlying scene and scenario construction, in-dependent of autobiographical content.

Importantly, our design did not include explicit instructions regarding the perspec-tive (first- or third-person) to be taken, nor did we assess the perspective adopted during imagery. We have clarified these points in the revised Discussion on p.33, l. 1093-110.

The manuscript states: “Extensive training was given to ensure that participants un-derstood these instructions correctly and were able to differentiate between the three imagery conditions.” It would be useful to describe in more detail how this training was conducted.

Response: We apologize for the lack of clarity. We have now included a more de-tailed description and replaced the broad term “extensive” training with the concrete training steps. See p. 9, l. 262-273.

Sample size: N = 19 is relatively small for an fMRI study, although power may be increased by the 7T resolution and PLS analysis. It would be useful to discuss the limits of generalizability and possible interindividual variability.

Response: We agree that discussing the implications for generalizability is crucial and have added a dedicated paragraph to address this limitation in our revised manuscript on p. 33, l. 1081-1092

Were participants excluded only in the “Eye-tracking data analysis”? The text re-ports that 22 data points were removed; it would be helpful to specify how many par-ticipants in total were excluded. In repeated-measures ANOVA, if a data point is re-moved, the entire participant must typically be excluded.

Response: Thank you for this important clarification request. You are correct to point out the potential confusion in our reporting. To clarify: the "22 data points" mentioned in the eye-tracking analysis refers to individual trial-level measurements that were removed as outliers (beyond 3 standard deviations), not participants. These removals occurred across the 1,140 total trials from all 19 participants in the final sample.

Regarding participant exclusions, three participants were excluded from the entire study (not just eye-tracking) due to suboptimal performance in the classification task during scanning, reducing our sample from 22 initially enrolled to 19 participants. No additional participants were excluded specifically from the eye-tracking analysis.

You raise an excellent methodological point about repeated-measures ANOVA and missing data. Traditional repeated-measures ANOVA does indeed typically require complete data, and missing data points often necessitate excluding entire participants to maintain a balanced design. However, o

---

## [Decision Letter · Decision Letter 1]

14 Dec 2025

Dear Dr. McCormick,

Thank you for submitting your manuscript to PLOS ONE. After careful consideration, we feel that it has merit but does not fully meet PLOS ONE’s publication criteria as it currently stands. Therefore, we invite you to submit a revised version of the manuscript that addresses the points raised during the review process.

We look forward to receiving your revised manuscript.

Kind regards,

Akitoshi Ogawa, Ph.D.

Academic Editor

PLOS One

Journal Requirements:

Additional Editor Comments:

The reviewers raised a few minor concerns. Can you address the concerns?

Reviewers' comments:

Reviewer's Responses to Questions

**Comments to the Author**

Reviewer #1: (No Response)

Reviewer #3: All comments have been addressed

2. Is the manuscript technically sound, and do the data support the conclusions?

Reviewer #1: Yes

Reviewer #3: Yes

3. Has the statistical analysis been performed appropriately and rigorously?

Reviewer #1: Yes

Reviewer #3: Yes

4. Have the authors made all data underlying the findings in their manuscript fully available?

Reviewer #1: Yes

Reviewer #3: Yes

5. Is the manuscript presented in an intelligible fashion and written in standard English?

Reviewer #1: Yes

Reviewer #3: Yes

Reviewer #1: I think the authors addressed my points satisfactorily, the paper has improved very much in clarity and balance. Very interesting data the fact that different parts of vmPFC dealt with scene and scenario construction.

Page 4, line 78: there is unfortunately a typo in German and the sentence is incomplete, so please correct that so that we can assess what is stated. Please also specify the type of patients included in that research. It now reads: "and found that both patient groups were impairedKlicken oder tippen Sie hier, um Text. "

Page 29: "Lesion studies have demonstrated that vmPFC patients, similar to hippocampal patients, show impairments in scene construction tasks, such as boundary extension, episodic future thinking and autobiographical memory tasks [7,8,15,54–56]". I think they refer to citation 16, not 15, as 15 is not about boundary extension.

Reviewer #3: After carefully reviewing the resubmission, I could assess that the authors have carefully and comprehensively addressed the concerns raised in the previous round of review. The revised manuscript is clearer in its conceptual framing, more transparent in its methodological description, and more cautious and precise in the interpretation of the findings. In particular, the distinction between object, scene, and scenario construction is now better articulated, and the role of the ventromedial prefrontal cortex is discussed in a more nuanced manner that aligns well with both the neuroimaging and neuropsychological literature. The additional clarifications regarding task instructions, training procedures, statistical analyses, and limitations substantially improve the rigor, readability, and reproducibility of the study. Therefore, I have now minor comments for the authors:

Comment #1: Despite improvements, the Results section remains dense due to the richness of analyses. I recommend adding brief subheadings or signposting sentences at the start of major Results subsections (e.g., “Behavioral results,” “Eye-tracking results,” “PLS task effects,” “PLS connectivity”).

Comment #2: The discussion of potential vmPFC subregional specialization is appropriate but somewhat speculative. Thus, I suggest authors add one qualifying sentence explicitly stating that the data do not allow strong claims about vmPFC subregional dissociations, and this remains a hypothesis for future high-resolution or lesion-informed work. This would further guard against overinterpretation.

**Do you want your identity to be public for this peer review?** For information about this choice, including consent withdrawal, please see our Privacy Policy

Reviewer #1: No

Reviewer #3: **Yes:** Sandra Carvalho

---

## [Author Response · Author response to Decision Letter 2]

5 Jan 2026

Dear Reviewers,

we thank the reviewers for their thoughtful and constructive feedback, which has greatly helped us to clarify and strengthen the manuscript. Below, we provide a point-by-point re-sponse to each of the reviewer’s comments. Reviewer comments are reproduced in black, followed by our detailed responses in blue. All major textual changes have been incorpo-rated into the revised version and are highlighted accordingly.

We hope that the revised version meets the reviewers’ expectations and provides a clearer account of our findings.

With kind regards

Cornelia McCormick

(on behalf of all co-authors)

Reviewer #1:

I think the authors addressed my points satisfactorily, the paper has improved very much in clarity and balance. Very interesting data the fact that different parts of vmPFC dealt with scene and scenario construction.

Page 4, line 78: there is unfortunately a typo in German and the sentence is incomplete, so please correct that so that we can assess what is stated. Please also specify the type of patients included in that research. It now reads: "and found that both patient groups were impairedKlicken oder tippen Sie hier, um Text. "

We thank the reviewer for pointing out the reference error. We have now corrected this and included the following paragraph:

“The authors also leveraged boundary extension, i.e. a cognitive process, in which viewers, after seeing a scene, automatically construct an internal representation that extends beyond the actual visual boundaries, leading to later memory for more than was shown, and found that both hippocampal and vmPFC lesions lead to impairments [16].”

Page 29: "Lesion studies have demonstrated that vmPFC patients, similar to hippocampal patients, show impairments in scene construction tasks, such as boundary extension, episodic future thinking and autobiographical memory tasks [7,8,15,54–56]". I think they refer to citation 16, not 15, as 15 is not about boundary extension.

We thank the reviewer for their careful evaluation of the references cited. We have now also included citation 16. We decided to keep citation 15 since it involves the comparison of vmPFC and hippocampal lesions:

“Lesion studies have demonstrated that vmPFC patients, similar to hippocampal patients, show impairments in scene construction tasks, such as boundary extension, episodic future thinking and autobiographical memory tasks [7,8,15,16,54–56].”

Reviewer #3:

After carefully reviewing the resubmission, I could assess that the authors have carefully and comprehensively addressed the concerns raised in the previous round of review. The revised manuscript is clearer in its conceptual framing, more transparent in its methodological description, and more cautious and precise in the interpretation of the findings. In particular, the distinction between object, scene, and scenario construction is now better articulated, and the role of the ventromedial prefrontal cortex is discussed in a more nuanced manner that aligns well with both the neuroimaging and neuropsychological literature. The additional clarifications regarding task instructions, training procedures, statistical analyses, and limitations substantially improve the rigor, readability, and reproducibility of the study. Therefore, I have now minor comments for the authors:

Comment #1: Despite improvements, the Results section remains dense due to the richness of analyses. I recommend adding brief subheadings or signposting sentences at the start of major Results subsections (e.g., “Behavioral results,” “Eye-tracking results,” “PLS task effects,” “PLS connectivity”).

We thank the reviewer for this helpful suggestion. Subheadings delineating the major Re-sults sections have been adapted for improved clarity. Please see the Results section.

Comment #2: The discussion of potential vmPFC subregional specialization is appropriate but somewhat speculative. Thus, I suggest authors add one qualifying sentence explicitly stating that the data do not allow strong claims about vmPFC subregional dissociations, and this remains a hypothesis for future high-resolution or lesion-informed work. This would further guard against overinterpretation.

We agree with the reviewer and have added a paragraph addressing the interpretational limitations of the qualitative vmPFC specialization results.

“However, interpretation of these results is limited by the study design, which precluded formal testing of the underlying hypotheses. These issues should be addressed in future studies.”

---

## [Editor Report · Decision Letter 2]

9 Jan 2026

Dear Dr. McCormick,

Thank you for submitting your manuscript to PLOS ONE. After careful consideration, we feel that it has merit but does not fully meet PLOS ONE’s publication criteria as it currently stands. Therefore, we invite you to submit a revised version of the manuscript that addresses the points raised during the review process.

We look forward to receiving your revised manuscript.

Kind regards,

Akitoshi Ogawa, Ph.D.

Academic Editor

PLOS One

Journal Requirements:

Additional Editor Comments:

Reviewer 1 raised a few minor concerns as outlined below. I would like the authors to address these before I make a decision on acceptance.

Reviewer 1

I think the authors addressed my points satisfactorily, the paper has improved very much in clarity and balance. Very interesting data the fact that different parts of vmPFC dealt with scene and scenario construction.

Page 4, line 78: there is unfortunately a typo in German and the sentence is incomplete, so please correct that so that we can assess what is stated. Please also specify the type of patients included in that research. It now reads: "and found that both patient groups were impairedKlicken oder tippen Sie hier, um Text. "

Page 29: "Lesion studies have demonstrated that vmPFC patients, similar to hippocampal patients, show impairments in scene construction tasks, such as boundary extension, episodic future thinking and autobiographical memory tasks [7,8,15,54–56]". I think they refer to citation 16, not 15, as 15 is not about boundary extension.

---

## [Author Response · Author response to Decision Letter 3]

15 Jan 2026

Dear Reviewers,

we thank the reviewers for their thoughtful and constructive feedback, which has greatly helped us to clarify and strengthen the manuscript. Below, we provide a point-by-point re-sponse to each of the reviewer’s comments. Reviewer comments are reproduced in black, followed by our detailed responses in blue. All major textual changes have been incorpo-rated into the revised version and are highlighted accordingly.

We hope that the revised version meets the reviewers’ expectations and provides a clearer account of our findings.

With kind regards

Cornelia McCormick

(on behalf of all co-authors)

Reviewer #1:

I think the authors addressed my points satisfactorily, the paper has improved very much in clarity and balance. Very interesting data the fact that different parts of vmPFC dealt with scene and scenario construction.

Page 4, line 78: there is unfortunately a typo in German and the sentence is incomplete, so please correct that so that we can assess what is stated. Please also specify the type of patients included in that research. It now reads: "and found that both patient groups were impairedKlicken oder tippen Sie hier, um Text. "

We thank the reviewer for pointing out the reference error. We have now corrected this and included the following paragraph:

“The authors also leveraged boundary extension, i.e. a cognitive process, in which viewers, after seeing a scene, automatically construct an internal representation that extends beyond the actual visual boundaries, leading to later memory for more than was shown, and found that both hippocampal and vmPFC lesions lead to impairments [16].”

Page 29: "Lesion studies have demonstrated that vmPFC patients, similar to hippocampal patients, show impairments in scene construction tasks, such as boundary extension, episodic future thinking and autobiographical memory tasks [7,8,15,54–56]". I think they refer to citation 16, not 15, as 15 is not about boundary extension.

We thank the reviewer for their careful evaluation of the references cited. We have now also included citation 16. We decided to keep citation 15 since it involves the comparison of vmPFC and hippocampal lesions during an event construction task :

“Lesion studies have demonstrated that vmPFC patients, similar to hippocampal patients, show impairments in scene construction tasks, such as boundary extension, episodic future thinking and autobiographical memory tasks [7,15,16,54–56].”

---

## [Editor Report · Decision Letter 3]

18 Jan 2026

From single scenes to extended scenarios: the role of the ventromedial prefrontal cortex in the construction of imagery-rich events

PONE-D-25-20610R3

Dear Dr. McCormick,

We’re pleased to inform you that your manuscript has been judged scientifically suitable for publication and will be formally accepted for publication once it meets all outstanding technical requirements.

Kind regards,

Akitoshi Ogawa, Ph.D.

Academic Editor

PLOS One
---

## [Editor Report · Acceptance letter]

PONE-D-25-20610R3

PLOS One

Dear Dr. McCormick,

I'm pleased to inform you that your manuscript has been deemed suitable for publication in PLOS One. Congratulations! Your manuscript is now being handed over to our production team.

Kind regards,

on behalf of

Dr. Akitoshi Ogawa

Academic Editor

PLOS One